# Revealing a novel nociceptive network that links the subthalamic nucleus to pain processing

Arnaud Pautrat[1,2], Marta Rolland[1,2], Margaux Barthelemy[1,2], Christelle Baunez[3], Valérie Sinniger[2,4], Brigitte Piallat[1,2], Marc Savasta[1,2], Paul G Overton[5], Olivier David[1,2], Veronique Coizet[1,2]*

[1]Inserm, Grenoble, France; [2]Grenoble Institute of Neurosciences, Université Grenoble Alpes, Grenoble, France; [3]Institut de Neurosciences de la Timone, Aix-Marseille Université, Marseille, France; [4]Service d'Hépato-Gastroentérologie, CHU Grenoble Alpes, Grenoble, France; [5]Department of Psychology, University of Sheffield, Sheffield, United Kingdom

**Abstract** Pain is a prevalent symptom of Parkinson's disease, and is effectively treated by deep brain stimulation of the subthalamic nucleus (STN). However, the link between pain and the STN remains unclear. In the present work, using in vivo electrophysiology in rats, we report that STN neurons exhibit complex tonic and phasic responses to noxious stimuli. We also show that nociception is altered following lesions of the STN, and characterize the role of the superior colliculus and the parabrachial nucleus in the transmission of nociceptive information to the STN, physiologically from both structures and anatomically in the case of the parabrachial nucleus. We show that STN nociceptive responses are abnormal in a rat model of PD, suggesting their dependence on the integrity of the nigrostriatal dopaminergic system. The STN-linked nociceptive network that we reveal is likely to be of considerable clinical importance in neurological diseases involving a dysfunction of the basal ganglia.
DOI: https://doi.org/10.7554/eLife.36607.001

*For correspondence:
veronique.coizet@univ-grenoble-alpes.fr

Competing interests: The authors declare that no competing interests exist.

## Introduction

Pain is highly prevalent in Parkinson's disease (PD) and includes primary symptoms assumed to originate from a dysfunction of the central nervous system. Patients describe bizarre and unexplained painful sensations such as burning, stabbing, aching, itching or tingling sensations, predominating on the more affected side (*Schestatsky et al., 2007*). These symptoms are not directly related to the pain caused by the motor symptoms (*Ha and Jankovic, 2012*). Sensitivity to noxious stimulation is also increased in patients with PD, with or without pain symptoms (*Berardelli et al., 2012*; *Brefel-Courbon et al., 2013*; *Tinazzi et al., 2008*), and their nociceptive threshold is altered (*Chudler and Dong, 1995*; *Conte et al., 2013*; *Djaldetti et al., 2004*). Although it is well known that PD affects the basal ganglia, there is to date no clear description of a link between the basal ganglia and the cerebral network involved in pain. Interestingly, deep brain stimulation of the subthalamic nucleus (STN-DBS) in PD, a valuable and effective therapeutic technique for motor symptoms (*Krack et al., 2003*; *Limousin et al., 1998*), has also been shown to reduce pain (*Cury et al., 2014*; *Hanagasi et al., 2011*; *Kim et al., 2008*; *Klingelhoefer et al., 2014*). By contrast, the effects of dopamine replacement therapy on the pain symptoms in PD are controversial, with some investigations reporting an improvement, no effect or an aggravation of pain symptoms or nociceptive thresholds (*Conte et al., 2013*; *Dellapina et al., 2011*). These variable results of dopamine replacement therapy indicate a role for other systems in the pain symptoms observed in PD. Importantly in

**eLife digest** Parkinson's disease is a condition affecting the human brain that becomes worse over time. The most common symptoms are tremors, muscle spasms and movements that are much slower than normal; all of which decrease an individual's quality of life. Although there is currently no cure, the brain structures involved in Parkinson's disease are known. These are collectively termed the basal ganglia, and are often targeted to treat the symptoms of Parkinson's disease. For example, electrically stimulating the subthalamic nucleus (STN), one part of the basal ganglia, reduces muscle tremors and stiffness.

Pain is another common symptom in Parkinson's disease. Patients often report strange burning or stabbing sensations with no obvious physical cause. They are also likely to be more sensitive to painful stimuli and have a lower pain threshold than normal. This suggested that the brain circuits that allow us to perceive and process pain could be somehow involved in Parkinson's disease. Indeed, stimulating the STN is known to relieve pain in Parkinson's disease, as well as the muscle symptoms, but exactly how the STN might link up with the brain's 'pain network' remains poorly understood. Pautrat et al. therefore set out to explore the connection between pain networks and the STN, and determine its potential role in Parkinson's disease.

First, the electrical activity of nerve cells in the STN of rats was measured, which revealed that these cells do respond to mildly painful sensations. Experiments using dyes to label cells in both the STN and brain structures known to transmit painful signals showed that the STN was indeed directly linked to the brain's pain network. Moreover, rats with a STN that did not work properly also responded abnormally to painful stimuli, confirming that the STN did indeed influence their perception of pain. Finally, Pautrat et al. repeated their measurements of electrical activity in the STN, this time using rats that lacked the same group of nerve cells affected in the basal ganglia of patients with Parkinson's disease. Such rats are commonly used to model the disease in laboratory experiments. In these rats, the STN cells responded very strongly to painful stimuli, suggesting that problems with the STN could be causing some of the pain symptoms in Parkinson's disease.

This work reveals a new role for the STN in controlling responses to pain, both in health and disease. Pautrat et al. hope that their results will inspire research into more effective treatments of nerve pain in both Parkinson's disease and other neurodegenerative conditions.

DOI: https://doi.org/10.7554/eLife.36607.002

the present context, it has been demonstrated that pain relief following STN-DBS is superior to that following dopaminergic treatment, further positioning STN as a crucial structure in pain symptoms in PD and their relief (*Sürücü et al., 2013*).

The mechanism by which STN-DBS improves pain in PD patients remains unclear, which raises a fundamental question about the link between the subthalamic nucleus (STN) and pain. Preliminary evidence suggests that noxious stimulation can modulate background activity in the STN, at least in the Parkinsonian brain (*Belasen et al., 2016*; *Heise et al., 2008*). As a consequence, the STN could be linked to a nociceptive network involved in the perception of noxious stimuli, although this has yet to be explored fully. Despite the classical description of a sensory territory in the STN (*Alexander et al., 1986*) and the functional impact that STN sensory responses could have on the basal ganglia (*Baunez et al., 2011*), there is a paucity of information in the literature regarding the type of sensory stimuli that activate this structure and the afferent sensory source(s) (*Coizet et al., 2009*; *Hammond et al., 1978*; *Matsumura et al., 1992*). Two major subcortical central targets for ascending nociceptive information from the spinal cord are potential relays for nociceptive information to the STN: the superior colliculus (SC) and the parabrachial nucleus (PBN) (*Hylden et al., 1989*; *Craig, 1995*; *Klop et al., 2005*; *McHaffie et al., 1989*).

We have shown that the SC, a highly conserved but evolutionarily ancient subcortical multi-sensory structure, directly projects to the STN and is the primary, if not exclusive, source of visual input into the STN (*Coizet et al., 2009*; *Tokuno et al., 1994*). This projection shows that subcortical hyper-direct pathways exist between brainstem sensorimotor structures and the STN which predate, from an evolutionary perspective, those from the cortex — thus reinforcing the position of the STN as a critical input structure processing short latency visual signals in the basal ganglia (*Baunez et al.,*

*2011*). The substantia nigra pars compacta (SNc) is also on the input side of the basal ganglia and we have shown that SNc receives nociceptive-related afferents from the PBN (*Coizet et al., 2010*), raising the possibility that the STN may also receive such inputs.

It is interesting to note that the STN is well known to project heavily to basal ganglia output structures such as the substantia nigra pars reticulata (SNr) (*Alexander et al., 1986*; *Gurney et al., 2001*), which in turn projects to the SC and PBN (*Schneider, 1986*; *Deniau and Chevalier, 1992*), linking the STN, SNr and SC/PBN anatomically. With the STN in a position to modulate a nociceptive network involving the SC/PBN, the elevated activity in the STN in Parkinsonism (*Bergman et al., 1994*; *Albin et al., 1995*) could underlie some of the unexplained pain symptoms in this disease, with STN-DBS acting (at least in part) locally to achieve its analgesic effects. Nociceptive processing in the STN would also be consistent with the nucleus's role as part of the brain's interrupt circuitry (*Jahanshahi et al., 2015*), terminating behaviors that achieve negative outcomes, of which pain is a clear example.

Therefore, the main objective of the present work was to characterize the link between STN and nociception, answering the following questions:

Does the STN process nociceptive information? We explored the possibility that noxious stimulation could induce nociceptive responses in the rat STN with in vivo electrophysiology.

Is the STN linked to a nociceptive network? We tested the potential role of the SC and PBN in the transmission of nociceptive information to the STN, both physiologically and, in the case of the PBN, anatomically.

Can manipulation of the integrity of the STN change nociceptive responses measured behaviorally? We tested STN-lesioned and sham-operated rats using a hotplate test.

Finally, to examine the hypothesis that STN dysfunction could underlie some of the pain symptoms observed in PD, we evaluated whether STN nociceptive responses were abnormal in a rat model of PD.

We present convergent evidence that the STN is functionally linked to a nociceptive network and that STN nociceptive responses are affected in Parkinsonism. The objectives above are summarized in *Figure 1*.

## Results

### Nociceptive responses in STN

#### STN neurons

A total of 98 cells were recorded across the STN (*Figure 2A*). The STN neurons sampled in the present study were characterized by a triphasic action potential in the majority of cases (n = 88, mean duration = 2.1 ± 0.06 ms), the remaining cells having a biphasic action potential (n = 10, mean duration 1.58 ± 0.12 ms) (*Figure 2B*). The STN cells had a mean baseline firing rate of 7.39 Hz (± 0.53 Hz) and exhibited various spontaneous patterns of activity such as an irregular pattern (n = 42, 43%), a regular pattern (n = 26, 27%) and a bursting pattern (n = 20, 20%) (*Figure 2C*). The remaining cells (n = 10, 10%) exhibited a mixture of these features. These electrophysiological characteristics are concordant with those reported elsewhere in the literature in anesthetized rats (*Hassani et al., 1996*; *Kreiss et al., 1996*; *Hamani et al., 2004*).

#### Phasic response

Following noxious stimulation performed on the contralateral hindpaw (120, 0.5 Hz), 19 STN cells remained unresponsive (19%) whereas 79 STN neurons (81%) exhibited a phasic response to the footshock with several patterns of response:

I. Monophasic excitation (n = 42): the response consisted of a monophasic excitation. This type of response could be subdivided into two categories according to their latencies and durations:
   1. Monophasic short/long (n = 20, *Figure 3Aa*): the cells exhibited a short-latency, long-duration excitation.
   2. Monophasic short/short (n = 17, *Figure 3Ab*): the cells exhibited a short-latency, short-duration excitation.

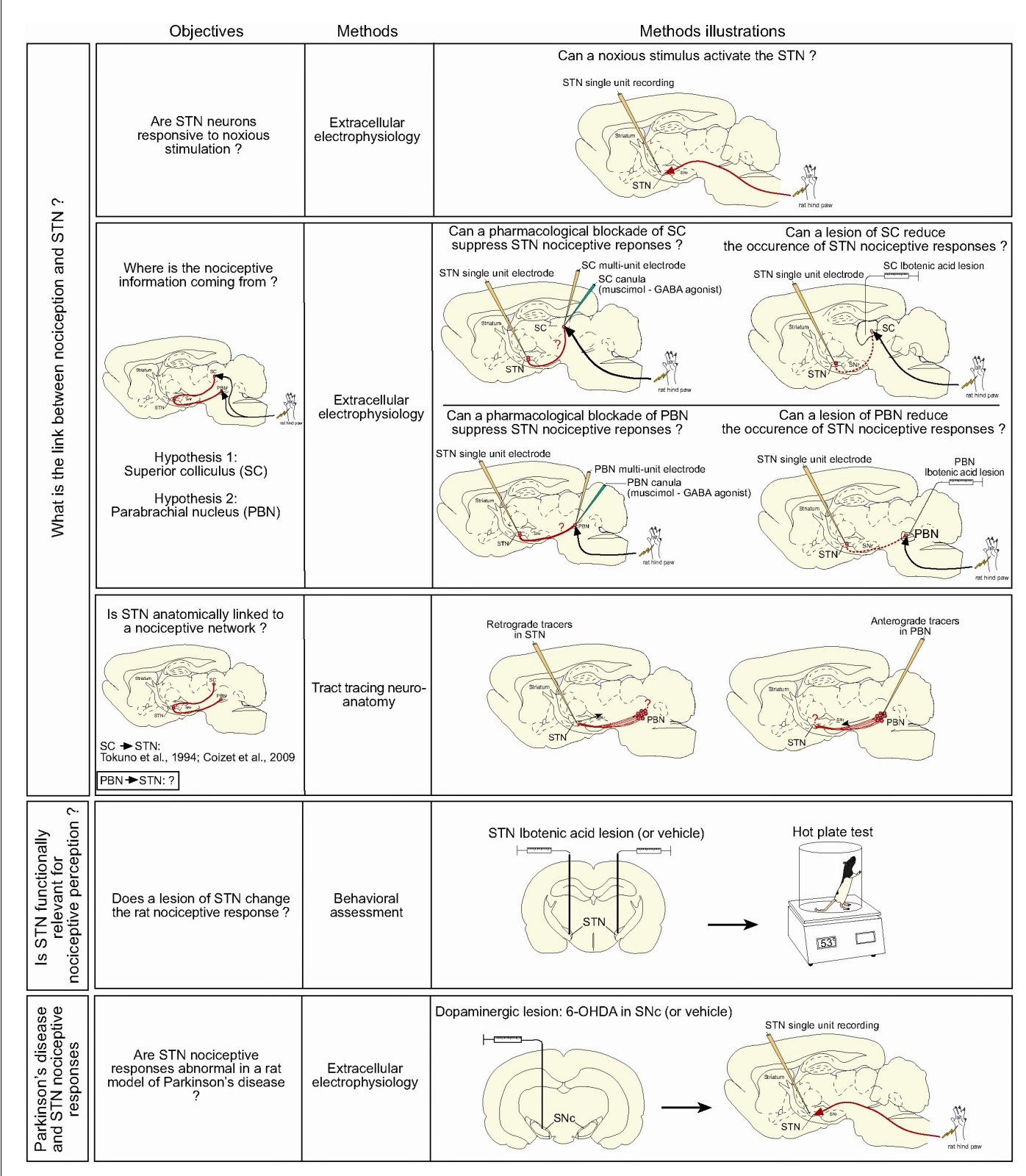

**Figure 1.** Summary of the objectives and methods.

DOI: https://doi.org/10.7554/eLife.36607.003

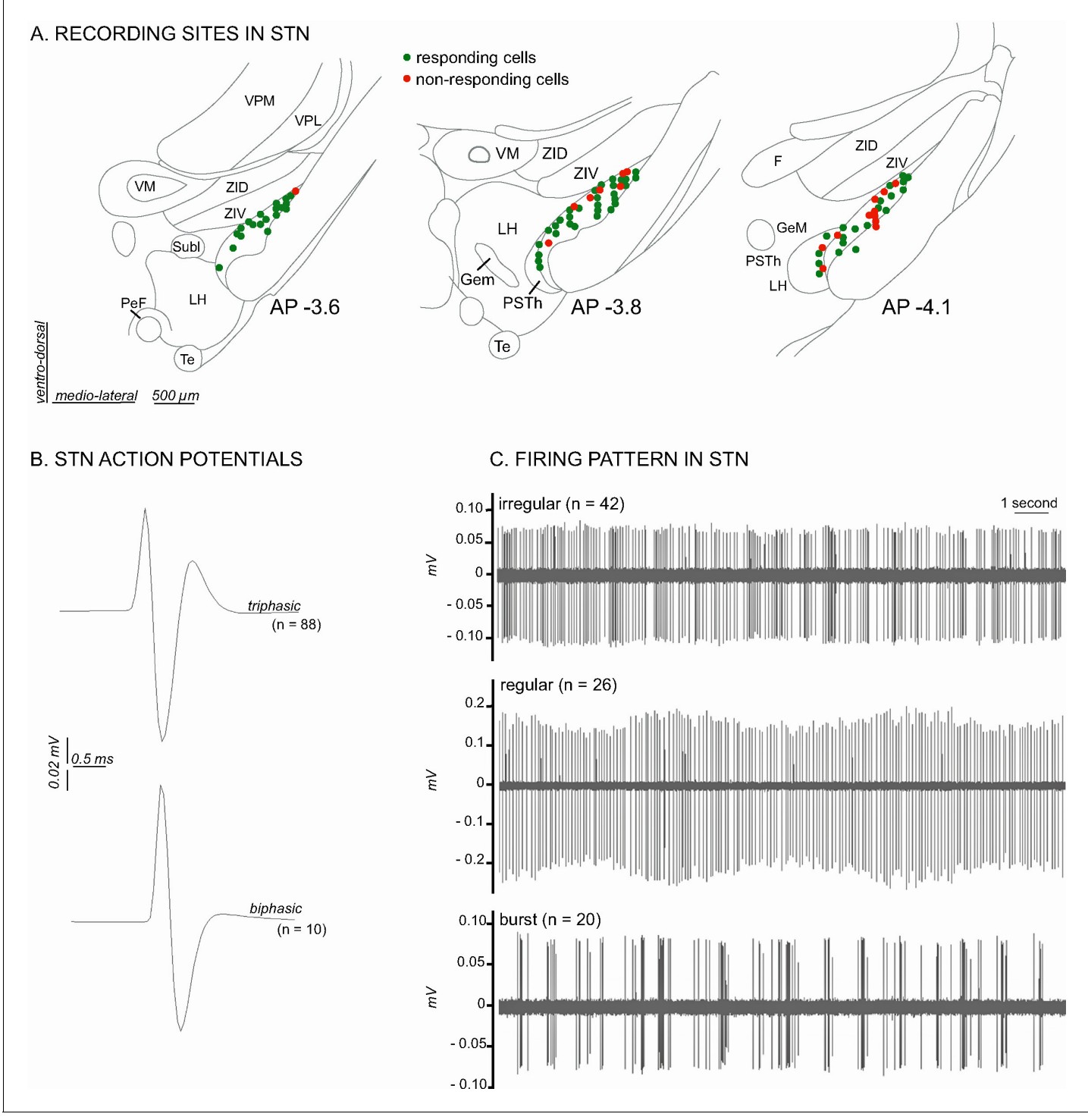

**Figure 2.** Histological and electrophysiological markers of recorded STN neurons. (**A**) Location of recording sites within the STN. Note that the number of non-responding cells is higher in the caudal part of the STN. (**B**) Example of triphasic (top) and biphasic (bottom) spike waveforms of STN neurons. (**C**) Individual recordings illustrating STN irregular (top), regular (middle) and in burst (bottom) firing patterns. Abbreviations: F, nucleus of the fields of Forel; GeM, Gemini hypothalamic nucleus; LH, lateral hypothalamic area; PeF, perifornical nucleus; PSTh, parasubthalamic nucleus; Subl, subincertal nucleus; Te, terete hypothalamic nucleus; VM, ventromedial thalamic nucleus; VPL, ventral posterolateral thalamic nucleus; VPM, ventral posteromedial thalamic nucleus; ZID, zona incerta, dorsal part; ZIV, zona incerta, ventral part.

DOI: https://doi.org/10.7554/eLife.36607.004

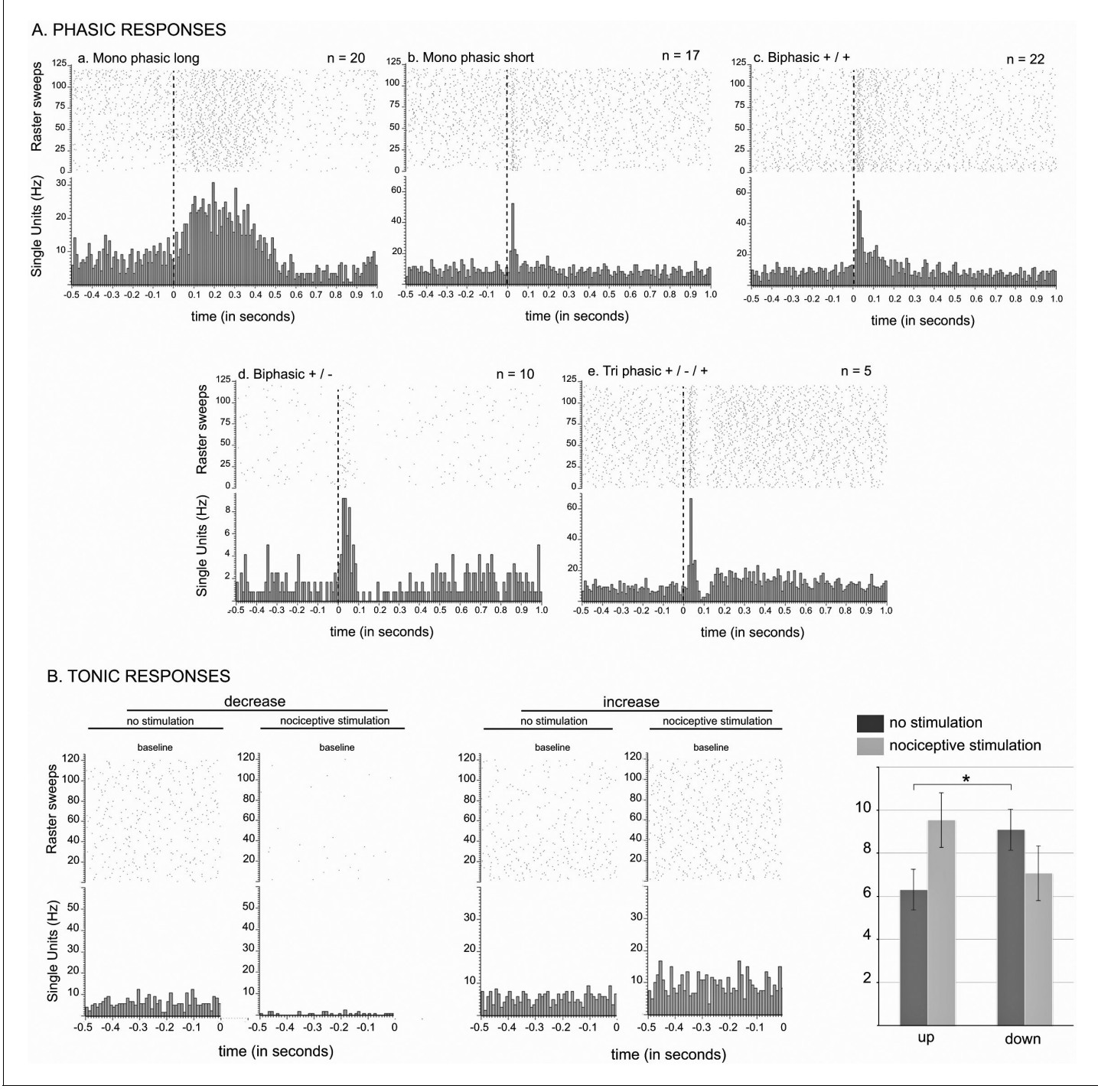

**Figure 3.** STN nociceptive phasic and tonic response. (**A**) Phasic responses. Peristimulus histograms showing individual cases of different types of phasic noxious-evoked responses in the STN. The dashed vertical line indicates the onset time of the nociceptive footshock. The n associated with each histogram indicates the number of cases exhibiting that class of response; total n = 98. (**B**) Tonic responses. Peristimulus histograms showing individual cases of decreased (left) and increased (right) STN baseline firing rate with the nociceptive stimulation. Histograms of the group mean data (±standard error of the mean [SEM]) during the sham (dark gray) and nociceptive (light gray) stimulation. The histograms show a significant increase or decrease of the baseline firing rate in the up (p<0.001) and down groups (p<0.001), respectively, and no effect of the no change group (p=0.1272). Note the higher baseline firing rate of the down cells during the sham stimulation compared to that of the up groups (p<0.05).
DOI: https://doi.org/10.7554/eLife.36607.005

II. <u>Biphasic +/+</u> (n = 22, *Figure 3Ac*): the cells had an initial short-latency, short-duration excitation followed by a longer-latency and longer-duration excitation.

III. <u>Biphasic +/-</u> (n = 10, *Figure 3Ad*): the response in these cells had two phases, a short-latency, short-duration excitation followed by an inhibition.

IV. <u>Triphasic +/-/+</u> (n = 5, *Figure 3Ae*): the response was characterized by an initial short-latency, short-duration excitation, then an inhibition or a marked reduction in firing rate followed by third late-latency and long-lasting excitation.

The remaining phasic responses could not be classified this way (n = 5).

The details of the latencies and durations of each response types can be found in *Table 1* below:

A significantly larger number of non-responding cells were located in the caudal portion of STN ($\chi2 = 6.94$, *df* = 2, p<0.05, *Figure 2A*). Of the 24 cells activated by noxious stimulation that were tested for multi-modal responses, only three exhibited an excitation in response to non-noxious somatosensory stimulation (light brush), hence the majority were nociceptive-only cells.

## Baseline firing rate

The introduction of noxious stimulation induced a statistically significant increase in STN baseline firing rate (Wilcoxon test: W[97] = −1199; p=0.05; mean ± SEM: no noxious stimulation: 7.13 ± 0.51 Hz vs noxious stimulation: 7.78 ± 0.52 Hz). However, we could clearly observe some cells whose baseline firing rate decreased when the footshock was delivered (*Figure 3B*). We therefore performed an individual analysis on each of the 98 STN cells, whether responding to the noxious stimulation of not, to test whether the change of their baseline firing rate after the introduction of the stimulation was statistically robust (Wilcoxon test, p<0.05), and if so, in which direction the change took place. We identified 39 (40%) and 17 (18%) cells showing a significant increase ('up' group) and decrease ('down' group), respectively, in their baseline firing rate with the stimulation, and no significant change for the remaining 42 cells (42%, 'no change' group). Contingency analysis did not reveal a specific topography of their location within STN, or a link to the shape of their action potential or to the presence or absence of a phasic response. Once grouped together in terms of direction, the 'up' and 'down' groups both exhibited a statistically significant change in their baseline firing rate. (Up — Wilcoxon test: W[38] = −780, p<0.001; mean ± SEM: no noxious stimulation 6.31 ± 0.83 Hz vs noxious stimulation 9.09 ± 0.94 Hz. Down — Wilcoxon test: W[16] = 153, p<0.001; mean ±SEM: no noxious stimulation 9.55. ± 1.43 Hz vs noxious stimulation 7.08 ± 1.26 Hz), unlike the 'no change' group (Wilcoxon test: W[41] = 245, p = 0.1272; mean ±SEM: no noxious stimulation 6.92 ± 0.65 Hz vs noxious stimulation 6.85 ± 0.67 Hz). Interestingly, the spontaneous firing of STN cells from the 'down' group had a significantly higher firing rate than that of the 'up' and 'no change' groups during the control period (Mann-Whitney test: U = 225, p<0.05; mean ± SEM: 'up' 6.31 ± 0.83 Hz vs 'down' 9.55 ± 1.43 Hz, U = 261, p=0.05; mean ± SEM: 'no change' 6.919 ± 0.65 Hz vs 'down' 9.55 ± 1.43 Hz), whereas the 'up' and 'no change' groups did not differ significantly. This suggests the presence of a separate population of STN neurons that have a higher firing rate.

## STN and nociceptive responses: is nociceptive information in the STN functionally relevant?

To evaluate the involvement of STN in nociceptive responses, we tested nociceptive responses in STN-lesioned and sham-operated rats behaviorally using a hot-plate test. The lesions were

**Table 1.** Response types latencies and durations

| RESPONSE TYPE | | PHASE 1 | | PHASE 2 | | PHASE 3 | |
|---|---|---|---|---|---|---|---|
| | | LATENCY | DURATION | LATENCY | DURATION | LATENCY | DURATION |
| I. | 1. | 37,80 ± 3,15 ms | 338.40 ± 49.79 ms | — | — | — | — |
| | 2. | 20.18 ± 3.15 ms | 34.47 ± 3.92 ms | — | — | — | — |
| II. | | 20.00 ± 3.02 ms | 35.00 ± 3.86 ms | 97.00 ± 6.97 ms | 269.00 ± 40.05 ms | — | — |
| III. | | 23.00 ± 4.61 ms | 69.00 ± 19.59 ms | 133.70 ± 32.76 ms | 176.70 ± 3.83 ms | — | — |
| IV. | | 16.4 ± 4.57 ms | 46.4 ± 7.90 ms | 101.2 ± 18.56 ms | 28.2 ± 5.27 ms | 227.6 ± 43.53 ms | 216.8 ± 69.3 ms |

DOI: https://doi.org/10.7554/eLife.36607.006

positioned within specific sub-regions of the STN, sparing the surrounding structures such as the zona incerta or the hypothalamus located above and medial to the STN, respectively. The bi-lateral STN lesions were localized in the posterior half (n = 6) or in the anterior/central (n = 6) parts of this structure, covering from 8% to 34% of the total surface of both bilateral STN (mean ± SEM: 20 ± 2.4%) (*Figure 4*).

These partial and highly localized STN lesions affected the nociceptive responses of the rats. Analysis showed a significant increase of the latency to produce the first sign of discomfort from the hotplate in the STN-lesioned group compared to the sham group (mean ± SEM: control = 10.81 ± 0.83 s; STN lesioned = 14.35 ± 0.9 s; p<0.05).

## Where does nociceptive information in the STN come from?

### Nociceptive responses in afferent structures

Footshocks produced short-latency, short-duration excitatory responses in the SC and PBN (*Table 2*). The PBN nociceptive responses were smaller in magnitude and amplitude than those in the SC. Footshocks did not change the spontaneous baseline firing rate in the SC or the PBN (SC: t [7] = 1.218; p=0.13; PBN: W = 8; number of pairs = 11; p=0.38). The latencies of the SC and PBN responses to the stimulation were both significantly shorter than those of the STN (SC-STN: t [15] = 2.88; p<0.001; mean ± SEM: SC 9 ± 0.8 ms vs STN 25.33 ± 5.27 ms; PBN-STN: t[22] = 3.34; p<0.01; mean ±SEM: PBN 11.55 ± 1.35 ms vs STN 24.54 ± 3.13 ms). Given that the response to noxious stimulation in SC and PBN occurs before that in the STN, both SC and PBN could be part of the nociceptive afferent network directed at the STN.

### Effect of SC or PBN inhibition on STN nociceptive responses

To test the possibility that the SC or PBN transmits nociceptive signals to the STN, we pharmacologically inhibited their neuronal activity with muscimol, a GABA$_A$ agonist, and evaluated the effect of their temporary inactivation on STN nociceptive responses. Simultaneous recordings were made from SC (multi-unit) and STN (single unit) neurons (n = 9) or from PBN (multi-unit) and STN (single unit) neurons (n = 13) before and after the delivery of a noxious footshock.

### Muscimol in the SC

The injection of muscimol adjacent to the SC electrode decreased the tonic activity in this structure (t[7] = 4.31; p<0.001) and abolished the phasic responses altogether in three cases. In the remaining cases, the muscimol produced a significant reduction of the magnitude (t[4] = 5.49; p<0.01) and the maximum amplitude (t[4] = 3.18; p<0.05), as well as a trend towards the duration of the response (t [4] = 1.74, p=0.08), but did not affect the latency (p=0.19). The depression of the SC neuronal activity by muscimol abolished STN nociceptive response in one case (1/9), and significantly reduced the duration of STN nociceptive responses in the remaining cases (t[7] )= 3.27; p<0.05), with a trend towards a decrease in magnitude, the difference being close to reaching the statistical threshold (t [7] = 2.31; p=0.054). The remaining parameters of the response and the baseline firing rate of STN cells were not statistically different to those in the pre-drug period.

### Muscimol in the PBN

The injection of muscimol adjacent to the PBN electrode significantly decreased the tonic activity of this structure (t[10] = 2.59; p<0.05) and abolished the phasic nociceptive responses in the PBN altogether in two cases. In the remaining cases, the injection of muscimol increased the latencies (t [8] = 3.12, p<0.01) and produced a significant reduction in the duration of the response (t[8] = 3.44, p<0.01), the magnitude of response (t[8] = 2.88; p<0.01) and the maximum amplitude (t[8] = 3.91; p<0.01). Unlike the SC, this general depression of PBN neuronal activity by muscimol completely abolished nociceptive phasic responses in five STN cells (5/13, *Figure 5*) and significantly reduced numerous parameters of the remaining STN responses to the stimulation, such as the duration (Wilcoxon test: W = 31, number of pairs = 8, p<0.05), magnitude (t[7] = 4.25; p<0.01) and maximum amplitude (t[7] = 3.21, p<0.01). Neither the latency (t[7] = 0.04, p=0.48, ns) nor the baseline firing rate (t[12] = 0.19, p=0.42, ns) were significantly affected by the injection of muscimol in PBN.

These results show that PBN pharmacological blockade with muscimol is more effective at reducing STN phasic nociceptive responses.

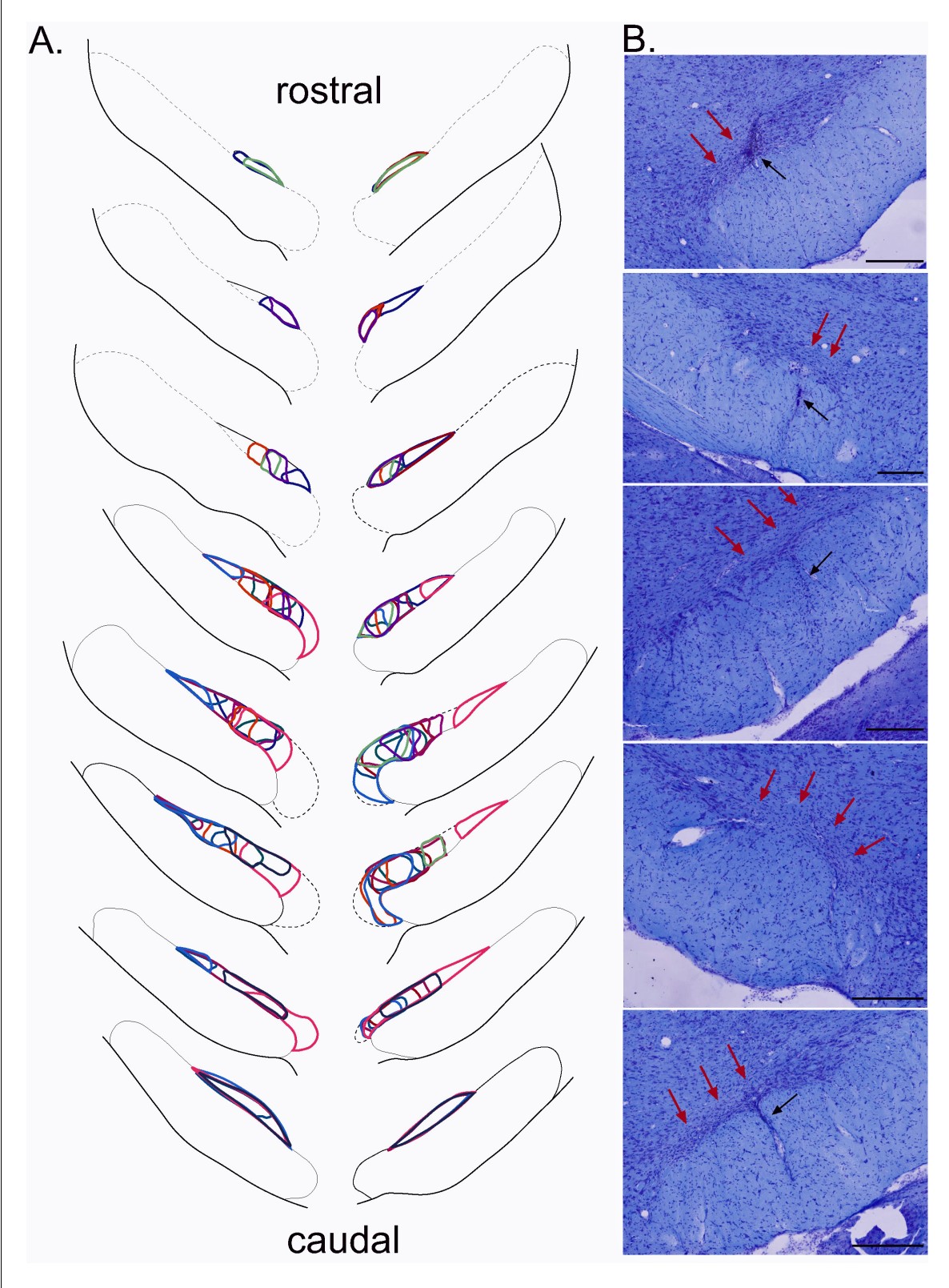

**Figure 4.** Plots of the STN lesion. (**A**) Schematic of the ibotenic acid STN lesions, each colour represents the plot of an individual animal. (**B**) Coronal sections (stained with cresyl violet) of the STN following a bilateral injection of ibotenic acid. Red arrows indicate the location of the lesion and black arrows the tract of the cannula. Scale bars = 400 μm.

DOI: https://doi.org/10.7554/eLife.36607.007

**Table 2.** Nociceptive responses in the subthalmic nucleus, superior colliculus and parabrachial nucleus.

| | Microinjection of muscimol in the superior colliculus | | | |
| | Superior colliculus | | Subthalamic nucleus | |
| | Pre muscimol | Post muscimol | Pre muscimol | Post muscimol |
|---|---|---|---|---|
| Latency | 9 ± 0.8 ms | 7.60 ± 0.68 ms | 25.33 ± 5.27 ms | 28.38 ± 7.75 ms |
| Duration | 19.62 ± 5.00 ms | 14 ± 3.03 ms | 176.13 ± 60.37 ms | 135.74 ± 50.19 ms * |
| Amplitude | 322.29 ± 58.16 Hz | 125.00 ± 26.22 Hz * | 15.61 ± 3.29 Hz | 14.4 ± 2.44 Hz |
| Magnitude | 140.20 ± 28.01 Hz | 45.01 ± 9.16 Hz * | 10.21 ± 1.99 Hz | 7.24 ± 1.41 Hz |
| Baseline FR no footshock | 18.27 ± 2.19 Hz | — | 4.37 ± 0.8 Hz | - |
| Baseline FR footshock | 20.79 ± 2.66 Hz | 7.69 ± 3.00 Hz * | 6.06 ± 1.48 Hz | 6.45 ± 1.38 Hz |
| | Microinjection of muscimol in the parabrachial nucleus | | | |
| | Parabrachial nucleus | | Subthalamic nucleus | |
| | Pre muscimol | Post muscimol | Pre muscimol | Post muscimol |
| Latency | 11.55 ± 1.35 ms | 19.00 ± 2.48 ms * | 24.54 ± 3.13 ms | 18.43 ± 1.73 ms |
| Duration | 26.45.12 ± 3.85 ms | 14 ± 3.49 ms * | 98.00 ± 27.35 ms | 45.75 ± 12.04 ms * |
| Amplitude | 146.97.42 ± 23.66 Hz | 83.26 ± 17.02 Hz * | 26.98 ± 5.72 Hz | 17.18 ± 3.26 Hz * |
| Magnitude | 45.16 ± 7.45 Hz | 23.70 ± 4.57 Hz * | 10.31 ± 2 Hz | 6.65 ± 1.44 Hz * |
| Baseline FR no footshock | 24.25 ± 1.97 Hz | — | 5.89 ± 0.49 Hz | — |
| Baseline FR footshock | 20.55 ± 2.42 Hz | 14.15 ± 2.76 Hz | 5.50 ± 0.71 Hz | 5.39 ± 0.57 Hz |

Mean ± SEM - * statistically different from pre-muscimol measure.

Abbreviations: cp, cerebral peduncle; IS, injection site; lPBN, lateral parabrachial nucleus; mPBN, medial parabrachial nucleus; scp, superior cerebral peduncle; STN, subthalamic nucleus.

Note: the amplitude of the response is the maximum amplitude during the response and the magnitude of the response is the mean number of single multi-unit events between response onset and offset minus the baseline mean.

DOI: https://doi.org/10.7554/eLife.36607.008

## Effect of SC or PBN lesions on STN nociceptive responses

Suppression of SC activity by muscimol only had a small influence on STN phasic nociceptive responses unlike PBN inactivation. However, while the technique of micro-injection offers a temporary inactivation with a possibility of recovery, the comparison of its effects on STN nociceptive responses versus the effects following PBN injections could be affected by the fact that muscimol was less effective at reducing SC phasic responses than those in the PBN, and by the difficulty of evaluating the spread of the injection in the SC and PBN. Therefore, we tested the effect of ipsilateral ibotenic acid lesions of the lateral part of the SC or PBN on STN phasic nociceptive responses.

## SC lesion

In all cases (n = 5), the lesion included the lateral SC and extended, in some cases, to the adjacent superficial and/or deep layers of this structure (*Figure 6A and B*). A total of 28 STN cells were recorded in lesioned animals before and after noxious footshocks. Of these cells, 27 (94%) still responded to the noxious stimulation. Analysis of the proportions of responding and non-responding cells comparing the control and SC-lesioned rats showed a significant difference between the two groups ($\chi2 = 2.98$, *df* = 1, p<0.05), with a larger proportion of responding cells in SC-lesioned rats compared to that in control animals (94% vs 81%). This correponded to a facilitation of the occurrence of STN nociceptive responses after the removal of the lateral SC.

Analysis of the nociceptive-induced phasic responses in STN showed a close to significant reduction of the duration (Mann-Whitney, U = 85, p=0.06), which is consistent with the reduction of duration observed previously after the injection of muscimol in SC. A plot of the distribution of STN responses according to their latency and duration in control and SC-lesioned animals shows that this statistical tendency could be due to the loss of the longer-lasting nociceptive responses in SC-lesioned animals (*Figure 6C*, dotted line box). This is supported by the observation of an increased proportion of short-latency/short-duration response types and a decrease in the proportion of long-

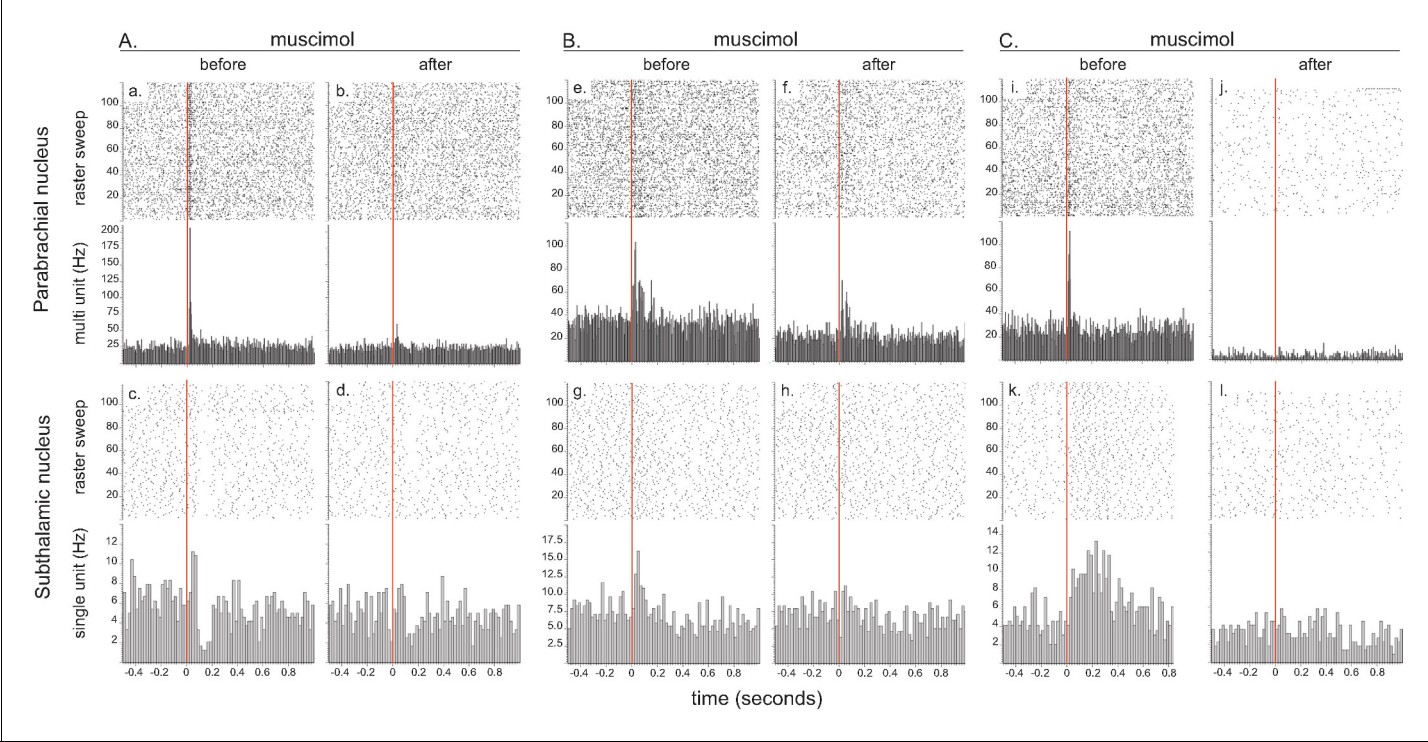

**Figure 5.** Effect of local injection of muscimol in PBN on STN nociceptive responses. The set of graphs presents the raster displays and peri-stimulus histograms of three single cases (**A, B and C**) aligned on the presentation of 120 stimuli (electrical footshocks delivered at 0.5 Hz; vertical red line). Prior to the injection of muscimol, both the PBN (**a, e, i**) and STN (**c, g, k**) neurons were responsive to the footshock. Following the injection of muscimol into the PBN, local neurons became less responsive (**b, f**) or unresponsive to the footshock (**j**) and so did the STN neurons (**d, h, l**). Note that PBN blockade with muscimol abolished different STN response types such as bipolar +/– (**a**), monophasic short/short (**g**) and monophasic short/long (**k**).
DOI: https://doi.org/10.7554/eLife.36607.009

lasting responses (*Figure 6D*). The other parameters of the response did not differ significantly and the firing rate was statistically unaffected.

## PBN lesion

The injection of ibotenic acid in the PBN induced a near total lesion of this structure in one case, extending over 87% of the PBN. This lesion also affected a small part of the caudal pedunculopontine nucleus. In the remaining cases, the lesion was only partial and affected between 30% to 60% of this nucleus (*Figure 7A and B*).

A total of 30 STN cells were recorded in PBN-lesioned rats before and after noxious footshocks. Although the lesion did not cover the entire PBN, the proportion of responding cells in the STN was significantly reduced, with 15 cells (50%) still responding to the noxious stimulation while the other 15 (50%) did not respond. Analysis of the proportions of responding and non-responding cells comparing the control and PBN lesion rats showed a significant difference between the two groups ($\chi 2 = 11.03$, *df* = 1, p<0.001), with a larger proportion of non-responding cells in PBN-lesioned rats than in control animals (50% vs 19%), representing a significant suppression of STN nociceptive responses after the removal of the PBN. Analysis of the nociceptive-induced phasic responses in the responding STN cells showed a significant reduction of response duration (Mann-Whitney, U = 378, p=0.05), which is consistent with the reduction of duration previously observed after the injection of muscimol in PBN. However, constrasting with the responses seen after the pharmacological blockade, the magnitude and maximum amplitude of the responses were not affected. As expected, the latency and the baseline firing rates were not statistically different in the PBN-lesioned and control rats.

A plot of the distribution of STN nociceptive responses according to their latency and duration in control and PBN-lesioned animals shows that most types of response described in control rats

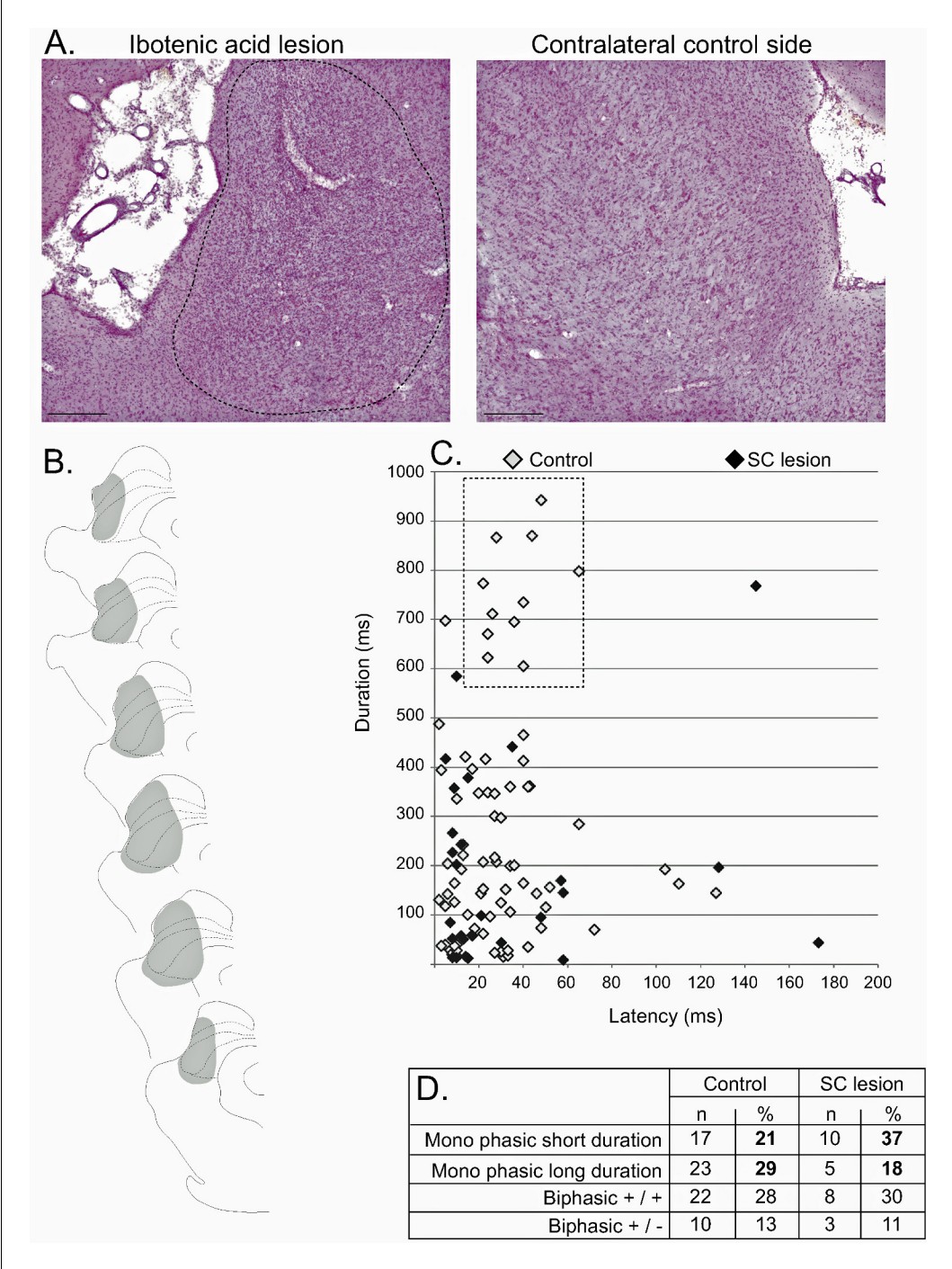

**Figure 6.** Effect of SC lesion on STN nociceptive responses. (**A**) Coronal sections (stained with cresyl violet) of the SC following a unilateral injection of ibotenic acid (dotted line, left) and its control contralateral side (right). Scale bars = 500 µm. (**B**) Schematic of a typical lesion (in gray) with ibotenic acid in the SC. (**C**) Plot of STN phasic noxious evoked responses in the STN according to their duration and latency in control (white) and SC lesioned (black) animals. The box outlined with a dotted line highlights the absence of long-duration responses in SC lesioned rats. (**D**) Table showing the proportions of different STN nociceptive response types in control and SC-lesioned animals. Note that the proportion of STN long-lasting responses decreases in SC-lesioned animals.

DOI: https://doi.org/10.7554/eLife.36607.010

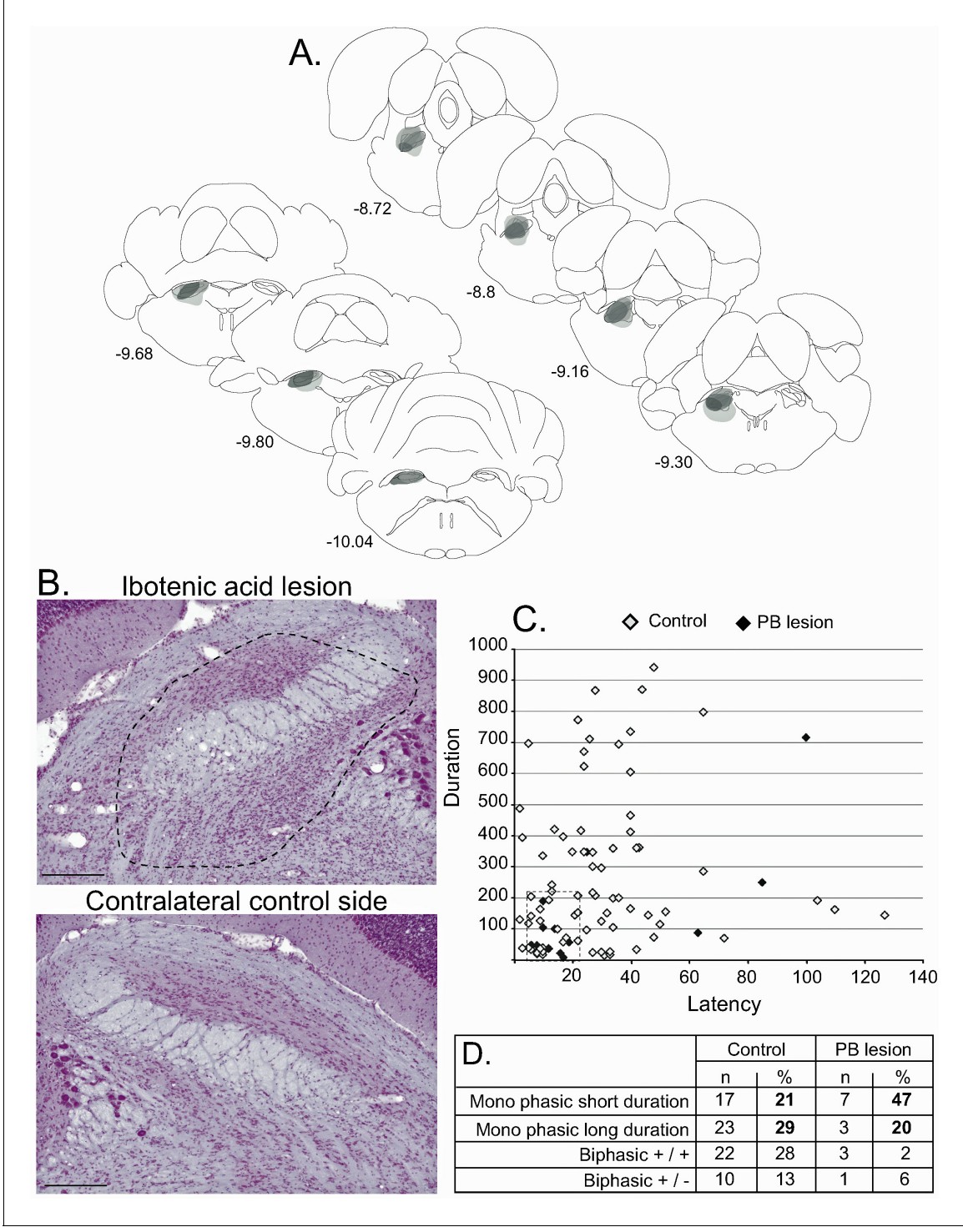

**Figure 7.** Effect of PBN lesion on STN nociceptive responses. (**A**) Schematic of the ibotenic acid lesion of the PBN. Each individual lesion is illustrated in a different tone of gray. (**B**) Coronal sections (stained with cresyl violet) of the parabrachial nucleus following a unilateral injection of ibotenic acid (dotted line, top) and its control contralateral side (bottom). Scale bars = 500 μm. (**C**) Plot of STN phasic noxious-evoked responses in the STN according to their duration and latency in control (white) and PBN-lesioned (black) animals. Note that PBN lesion abolished all types of STN nociceptive response. (**D**) Table showing the proportion of STN nociceptive response types in control and PBN-lesioned animals.

DOI: https://doi.org/10.7554/eLife.36607.011

disappeared. A cluster of responding cells with short latencies and durations remained (*Figure 7C and D*).

## Anatomical link between the PBN and the STN

Our electrophysiological experiments suggest that the PBN acts as a critical source of nociceptive input into the STN and the SC as a critical modulator of those responses. While the connection between the SC and the STN has been characterized previously (*Coizet et al., 2009*; *Tokuno et al., 1994*), the functional link between the PBN and the STN was puzzling as a previous neuroanatomical study characterizing afferent connections of the STN reported the absence of labeled terminals and fibers in the STN following the injection of an anterograde tracer in the parabrachial complex (*Canteras et al., 1990*). The characterization of PBN efferent connections has often focused on the amygdala or the hypothalamus, but a close examination of PBN-related fiber pathways, especially to the amygdala, indicates that those fibers are travelling close to (*Saper and Loewy [1980]*: *Figure 4F*) or even partly within the STN (e.g. *Bernard et al. [1989]*: *Figure 2J*; *Sarhan et al. [2005]*: Figure 9B3). Therefore, to better understand the anatomical basis of our electrophysiological data, we re-evaluated the parabrachio-subthalamic pathway using a combination of anterograde and retrograde tract-tracing techniques.

## Anterograde tract-tracing

Injections of the anterograde tracers *Phaseolus vulgaris* leucoagglutinin (PHA-l, n = 4) or biotinylated dextran amine (BDA, n = 4) into the PBN revealed a robust direct projection to the ipsilateral STN and a less substantial projection to the contralateral STN. The ipsilateral ascending fibers leave the PBN in an antero-dorsal direction and pass above and through the dorso-caudal pedunculopontine tegmental nucleus (*Figure 8*). There, they split into three large pathways (*Figure 8A*), a PBN-SC (intermediate and deep layers), a PBN-thalamic and a PBN-nigral ventral projection. PBN-labeled axons traveling to the STN are a rostral extension of the pathway we have described previously (*Coizet et al., 2010*) from the PBN to the dopaminergic neurons in the ventral midbrain, the parabrachio-nigral pathway. A substantial number of fibers continue further forward to the amygdala and the cortex (*Figure 8A*). This parabrachio-subthalamic pathway originates in the lateral and medial PBN (lPBN and mPBN, respectively). Thus, injections of either PHA-l or BDA centered preferentially on the lPBN or the mPBN were both associated with numerous labeled axons and terminals, which were differentially distributed within sub-regions of the STN. PBN fibers and terminals were largely seen in a dorsal sheet that extended across the entire mediolateral axis of the STN (*Figure 8B and C*). Moving rostrally, they further spread across the dorsoventral area of the STN. Many anterogradely labeled boutons were located in the vicinity of labeled fibers (*Figure 8B and C*).

*Retrograde track tracing.* To identify the regional distribution and morphology of the parabrachio-subthalamic projection neurons, we injected small quantities of the retrograde tracers Fluorogold (n = 4) or Cholera toxin subunit B (CTB, n = 2) into the STN. The projection to the STN exhibited little topography. Retrogradely labeled neurons were found in all subnuclei of both the ipsilateral and contralateral PBN, and also within the fibers of the superior cerebellar peduncle (cp) (*Figure 9A and B*). The density of the labeled cells was however significantly larger on the ipsilateral side, as confirmed by a three factor (Side: ipsilateral/contralateral; Level: AP 8.8/AP 9.3/AP 9.8; Subdivisions: lateral/medial/cp) repeated-measures ANOVA (Side factor: F = 9.33, $df$ = 1, p<0.05), and varied within the PBN according to the AP level (interaction between the level and the subdivisions, F = 4.10, $df$ = 4, p<0.05). The density of the cells increased in the lateral PBN when moving rostrally and the majority of the cells in the mPBN were located in the posterior PBN. PBN neurons that were retrogradelly labeled by STN injections were small to medium-sized (mean ± SEM: 229.25 ± 8.21 $\mu m^2$, range from 102.1 to 681.10 $\mu m^2$; n = 184) with a majority of round (59%) and bipolar (35%) soma and some multipolar cell bodies (6%) (*Figure 9C*).

## STN nociceptive responses in a rat model of Parkinson's disease

In a classical rat model of PD induced by an injection of 6-hydroxydopamine (6-OHDA), a neurotoxin thattargets dopaminergic neurons (DA), in the SNc, we tested whether STN nociceptive responses were dysfunctional.

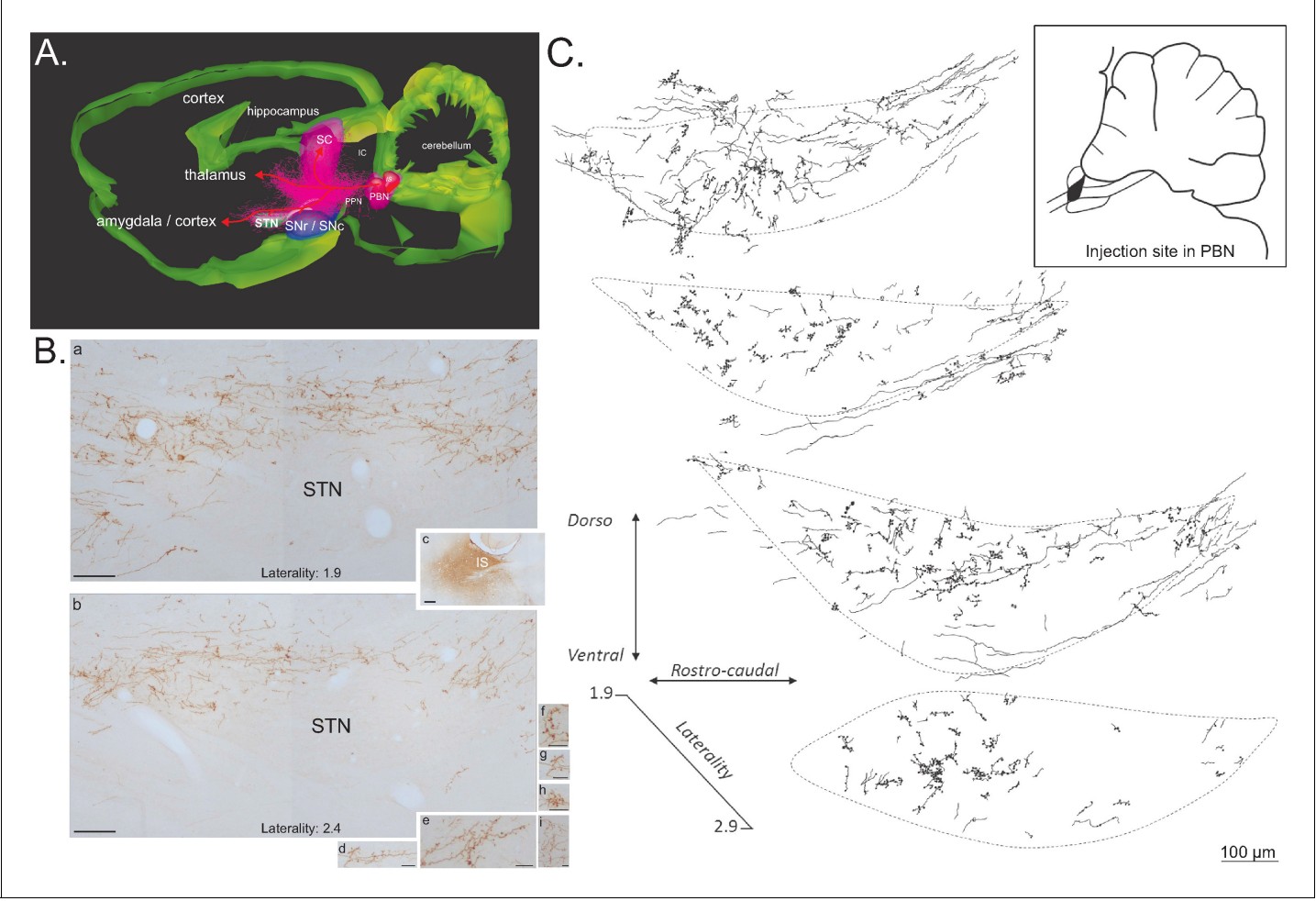

**Figure 8.** Anterograde tracer in the PBN. (**A**) 3D renderings of parasagittal brain sections covering the PBN and STN width, illustrating the different bundles leaving the PBN following a local injection of an anterograde tracer (PHAL). (**B**) Sagittal sections illustrating a PHAL injection site in the lateral PBN (**c**), associated with labeled terminals in the medial (**a**) and lateral (**b**) STN. PBN labeled terminals contain dense synaptic bouton mainly localized in STN dorsal sector (**d–i**). Scale bars: a–c = 200 μm, d–i = 20 μm. (**C**) Schematic illustrating the location of terminals and synaptic boutons in the STN following the injection of biotinylated dextran amine (BDA) in the lateral portion of the rostral PBN (insert box). Abbreviations: IC, inferior colliculus; IS, injection site; PBN, parabrachial nucleus; PPN, pedunculopontin nucleus; SC, superior colliculus; SNc, substantia nigra pars compacta; SNr, substantia nigra pars reticulate; STN, subthalamic nucleus.

DOI: https://doi.org/10.7554/eLife.36607.012

TH immunohistochemistry was used to assess the extent of the dopamine denervation induced by 6-OHDA in the DA-lesioned rats. TH-labeled neurons on the lesioned side were reduced to an average of 6.13 ± 0.71% (mean ± SEM) of those on the unlesioned side, with the remaining neurons located in the medial part of the SNc. The reduced number of dopaminergic cells led to an average decrease of 65.22 ± 2.08% in the dopaminergic innervation of the striatum.

Totals of 34 and 43 cells were recorded in the control and DA-lesioned rats, respectively. As expected in a model of PD, STN cells had a significantly higher firing rate in DA-lesioned rats compared to controls (mean ± SEM: control = 8.00 ± 1.04 ms; DA-lesioned = 12.96 ± 2.02 ms; p<0.05, *Figure 3A*). STN responses to nociceptive stimulation were abnormal in the DA-lesioned group. Analysis revealed that STN cells in PD rats exhibited significantly longer responses (mean ± SEM: control = 84.39 ± 16.75 ms; DA-lesioned = 175.38 ± 39.85 ms, p<0.05) with a greater amplitude (mean ± SEM: control = 27.81 ± 3.58 ms; DA-lesioned = 40.88 ± 5.91 ms; p<0.05, *Figure 3C*) and magnitude (mean ± SEM: control = 8.13 ± 1.38 ms; DA-lesioned = 13.11 ± 2.47 ms;p<0.05) compared to the sham control animals. The proportion of cells exhibiting the three levels of STN

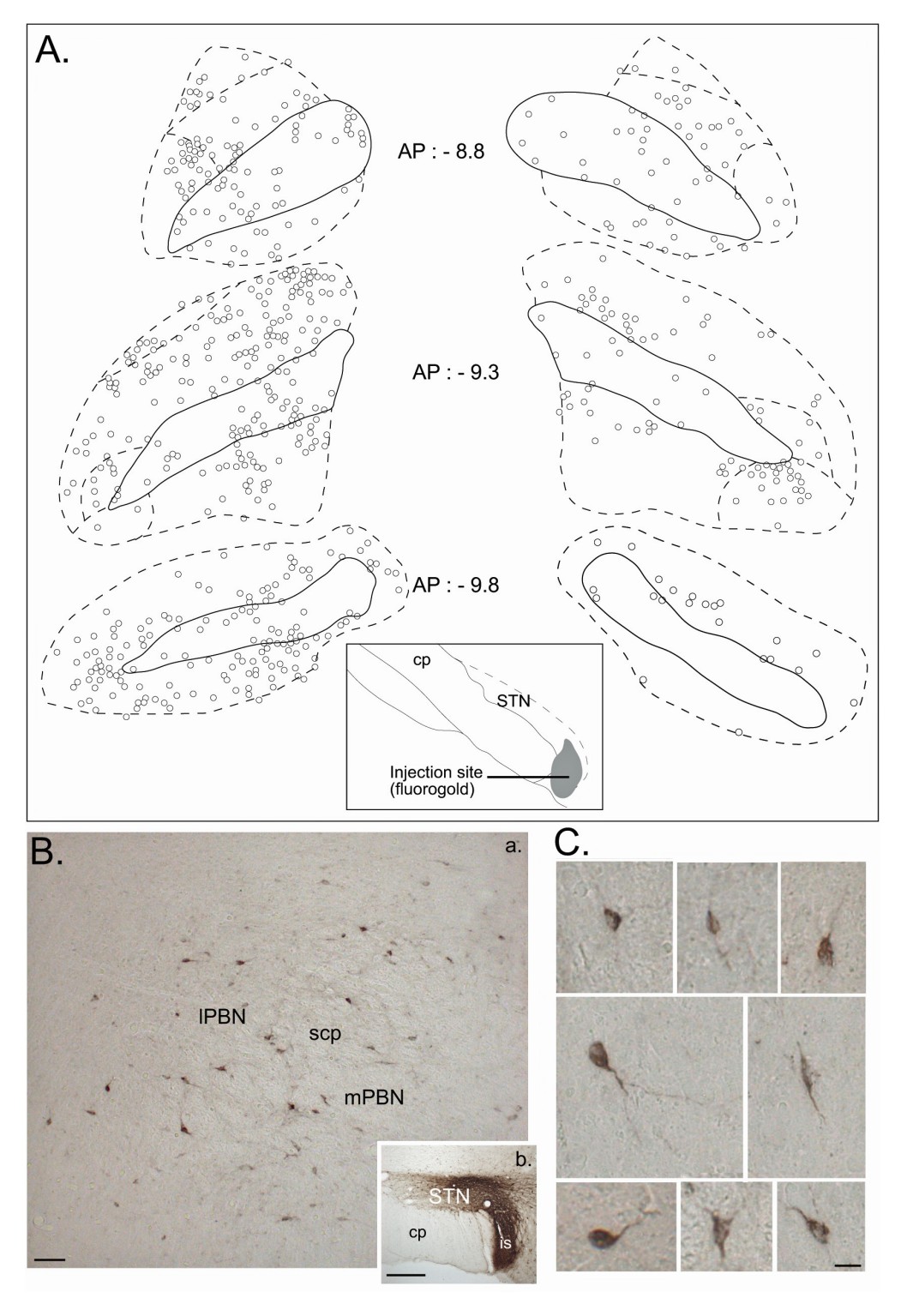

**Figure 9.** Retrograde tracer in the STN. (**A**) Drawing of coronal sections centered on the ipsilateral (left) and contralateral (right) PBN illustrating the location of the retrogradely labeled cells following an injection of a retrograde tracer Fluorogold in the STN. (**B**) Photomicrographs of retrogradely labeled neurons in the PBN (**a**) following the injection of cholera toxin unit B (CTB) in the STN (**b**). Scale bars = 200 µm. (**C**) Morphology of retrogradely labeled PBN neurons following an injection of CTB into the STN. Scale bars: 20 µm.

DOI: https://doi.org/10.7554/eLife.36607.013

baseline firing rate upon introduction of the stimulation (up, down or no change) was not altered in the PD rat groups ($\chi 2 = 0.32$; p=0.85). Therefore, STN phasic nociceptive responses in PD rats were exacerbated, while the tonic modulation of the firing rate was preserved.

## Discussion

The present study demonstrates for the first time that a large majority of STN neurons exhibit various mono- or multi-phasic responses to noxious stimulation, consistent with the hypothesis that one of the functions of the STN is to interrupt behavior when appropriate (*Jahanshahi et al., 2015*); in this case, to select a more appropriate action to try to relieve the noxious sensation. STN nociceptive responses mainly had a short latency (~20–40 ms) and could be recorded all over the structure. We have shown that most of the responsive cells are nociceptive-specific, as only few of them also respond to non-noxious somatosensory stimulation. In addition, we found that we could differentiate three types of STN neurons, which showed an increase, a decrease or no change, respectively, in their baseline firing rate upon introduction of the noxious stimulation. The spontaneous firing rate of STN 'down' cells was significantly higher than that of the two other groups, suggesting the possibility of a separate group of cells. Furthermore, we have shown that STN responses to nociceptive stimuli were abnormal in a rat model of PD, suggesting that the nociceptive responses recorded in STN depend on the integrity of the nigrostriatal DA system.

When determining the afferent source of nociception-related influence on STN activity, we have revealed a crucial role for two brainstem structures, the PBN and the SC, by demonstrating the effects of their inactivation on nociceptive responses in the STN and also by highlighting the existence of an anatomical direct pathway from the PBN to the STN. This parabrachio-subthalamic projection represents a second example of a subcortical hyperdirect pathway to the STN from a sensorimotor structure, in addition to the tecto-subthalamic pathway described previously (*Coizet et al., 2009*; *Tokuno et al., 1994*). Finally, we have shown that these anatomico-electrophysiological findings translate into a functional role for the STN in mediating nociception, in that nociceptive behavioral responses were affected by lesions of the STN. However, a note of caution is required since we used only male rats, and thus care should be taken in extrapolating our results to females.

Using noxious electrical stimulation of the hindpaw allowed us to record precisely the timing of responses in our structures of interest with controlled parameters of stimulation. It is interesting to note that the majority of STN nociceptive responses had short latencies (76/79) and were monophasic (40/79), a pattern of responses that is similar in proportion to the pattern of STN responses following visual stimulation (*Coizet et al., 2009*) but dissimilar to the pattern of STN responses following stimulation of the frontal (*Magill et al., 2004*), sensorimotor (*Fujimoto and Kita, 1993*) or motor cortex (*Kolomiets et al., 2001*). The latter — in the majority of cases — are multi-phasic, with two excitatory phases (equivalent to the present biphasic +/+) often separated by an inhibition (equivalent to the present triphasic +/–/+). It has been hypothesized previously (*Magill et al., 2004*; *Kitai and Deniau, 1981*) that the short latency excitation following cortical stimulation appears to be driven by a hyperdirect pathway to the STN, whereas the later phases of the response arise from polysynaptic interactions, which are manifested more slowly. The inhibition following the first excitation has been hypothesized to involve the reciprocally connected STN–globus pallidus (GP) network (*Fujimoto and Kita, 1993*). STN excitation may activate GP GABAergic neurons, which in return inhibit the STN. Overall, our data suggest that when the rat is subjected to noxious stimulation, the main pathway to the STN that is activated is a fast hyperdirect pathway, originating in part in the PBN. The fact that we only have a few cells showing an inhibitory second phase (10 biphasic and 5 triphasic, 15/79) indicates that STN nociceptive cells are rarely closely connected to the GP and lack GP-STN feedforward control.

Functionally antagonistic STN neuronal subpopulations have been found in the STN. Specific GO and STOP cells have been described in PD patients performing a stop signal paradigm, during motor execution or response inhibition, respectively (*Benis et al., 2016*). Sub-populations of STN cells have also been shown to code exclusively for reward magnitude (4% vs 32% sucrose [*Lardeux et al., 2009*]), error-related activity ('Oops neurons' [*Lardeux et al., 2009*]), reward value (cocaine or sucrose [*Lardeux et al., 2013*]) and for positive and aversive reinforcers (*Breysse et al., 2015*).

A major finding of our work is that we were able to differentiate a subpopulation of STN neurons that had a higher spontaneous firing rate on the basis of the effect of noxious stimuli on general tonic activity. This finding is important as glutamatergic tone from the STN is likely to have a strong impact on the tonic level of activity in the basal ganglia network, especially in the output structures (as we hypothesized that nociceptive-responding STN cells may not be densely connected to GP). The results from previous computational studies (*Gurney et al., 2001*) suggest that tonic control by the STN may adjust the general level of activity of the inhibitory GABAergic neurons of the basal ganglia output structures, which are known to project to the SC and PBN (*Schneider, 1986*; *Deniau and Chevalier, 1992*). This tonic STN control is hypothesized to be used by the basal ganglia to optimize selection of the most appropriate action. The identification of separate subpopulations of cells in the STN according to their spontaneous firing rate, and the orientation of the change of their firing rate following the occurrence of noxious stimulation,suggests that the tonic excitatory effects of the STN may not be uniform, although further work is required to elucidate the connectivity of the STN subpopulations. This mechanism is important in the context of PD in which STN activity is pathologically increased (*Bergman et al., 1994*; *Albin et al., 1995*), probably disrupting this control. This possibility is further supported by our results showing enhanced phasic nociceptive responses in a PD rat model, with an increase in the latency to make nocifensive responses in the hotplate test following lesions of the STN. As well as interfering with action selection, disrupted STN control in PD would probably have an impact on the SC and PBN and their role in sensory signal processing.

In addition to demonstrating that STN neurons process nociceptive information, we also assessed whether two subcortical sensori-motor structures from the brain stem transmit nociceptive signals to the STN. Despite SC nociceptive responses having shorter latencies than those of STN neurons to the same stimulus, chemical suppression and lesions of SC had relatively minor effects on the responses of STN to noxious footshock. Lesions of the SC with acid ibotenic reduced the number of cells that do not respond to noxious stimulation, suggesting that the SC gates the pool of responding cells in the STN. Our previous work has demonstrated that the SC is a critical relay for short-latency visual input into DA neurons (*Dommett et al., 2005*) but not for short-latency nociceptive input (*Coizet et al., 2006*) transmitted by the PBN (*Coizet et al., 2010*). The current results suggest the same organization when considering visual and nociceptive input into the STN. The SC is a crucial structure to transmit visual information while the PBN strongly contributes to the relay of nociceptive signals. PBN lesions significantly reduced the number of STN cells responding to noxious stimuli, sparing a group of cells with short latency short duration responses (*Figure 7C*), which are possibly activated by nociceptive information relayed by the thalamus (*Dostrovsky, 2000a*). This nociceptive network linked to the STN is the probable substrate underlying the successful analgesic effects of STN deep-brain stimulation. *Kim et al. (2012)* hypothesized that STN-DBS improves secondary pain symptoms in PD because this stimulation decreases the abnormally increased muscle tone in patients and may alleviate the primary nociception processing in the central nervous system. DBS effects are complex and despite the success of DBS in treating a variety of psychiatric and neurological disorders, the mechanisms underpinning its therapeutic efficacy remain unclear (*McIntyre et al., 2004*; *Ashkan et al., 2017*). DBS is hypothesized to induce a 'functional lesion' of the STN (*Follett, 2000*), via depolarization blockade and synaptic inhibition (*Beurrier et al., 2001*; *Dostrovsky et al., 2000b*), which would lead to a suppression of the activity of STN neurons. We hypothesize that these mechanisms would reduce the pathologically increased firing rate in the STN in PD (and thus the pain symptoms), as well as nociceptive responses.

Our current work using anterograde tract-tracing neuroanatomy coupled with 3D reconstruction indicates that a small ascending bundle leaves the PBN and then splits into three massive projections traveling toward the SC, the thalamus and the SNc/STN. Some fibers from this last ascending pathway continue rostrally to terminate in the amygdala and the cortex. Comparison of the size of the bundle leaving the PBN and the size of the bundles traveling to their targets indicate that the number of labelled fibers clearly increase, suggesting that PBN axons have collaterals. This PBN-STN projection is possibly interconnected with other PBN efferents, such as the PBN–amygdala projection, which partly travels through the STN and has cells of origin that are of a similar type to those of the PBN cells projecting to STN (*Sarhan et al., 2005*). With a DBS effect on axons and fibers (*Chiken and Nambu, 2014*), the characterization of this projection and network are important in the context of the effects of STN-DBS on pain symptoms. Overall, STN-DBS would not only impact STN

and PBN nociceptive processing but would also modulate PBN-amygdala fibers and possibly other PBN efferents via the collaterals. The effect of STN-DBS would therefore impact many aspects of pain such as, for example, pain-related emotional reactions when activating the PBN-amygdaloid connection or neuroendocrine homeostatic regulation in response to pain by activating the PBN-hypothalamic pathway (*Gauriau and Bernard, 2002*). Further experiments are now needed to fully characterize the effect of STN-DBS on nociceptive processing in our rat models and how aspects of that network are modulated to achieve a DBS-related analgesic effect.

Pain is a multifaceted experience that can be understood in terms of somatosensory, affective and cognitive dimensions. DBS therapies that are focused on a single facet of pain, originally targeting somatosensory networks or more recently targeting affective regions (*Schneider, 1986*). The STN is a small structure with functional territories such as the limbic, cognitive and sensory, in close proximity to each other. This would allow the potential modulation of different modalities of pain and, in the future, the best placements of DBS electrodes within those territories would have to be tested to maximize the analgesic effect. Finally, numerous non-motor symptoms can worsen or improve depending on the electrical stimulation parameters, as well as the location of the electrode (*Kim et al., 2015*). The best parameters of stimulation for nociception would need to take into account the effect of those parameters on other symptoms of PD.

Finally, non-neuropathic pain, recently recognized as a frequent and disabling symptom in PD, is a complaint affecting many patients who have numerous neurodegenerative disease such as Alzheimer's disease and other dementias, motor neuron disease, Huntington's disease, spinocerebellar ataxia and spinal muscular atrophy (*de Tommaso et al., 2016*). Our findings on the involvement of the STN in nociceptive processing and its link to a nociceptive network open a new direction for research to explore a possible role of this structure in other pain syndromes, especially extra-pyramidal ones like Huntington's disease, which is characterized by a dysfunction of the basal ganglia. It also opens up the possibility of developing therapeutic strategies using DBS. A variety of brain sites have been identified for chronic stimulation procedures to attenuate pain (*Davis et al., 1998*). These targets include the thalamus, the periventricular gray nucleus, the cingulate cortex and the motor cortex (*Gorecki et al., 1989*; *Davis et al., 1998*). With the involvement of the STN in a nociceptive network as demonstrated in our work, the STN-DBS technique can thus be considered in the future as a new target for the treatment of pain not only in pharmaco-resistant patients suffering from previously described neurodegenerative disease, but also, for example, in those with chronic pain disease or pharmaco-resistant patients with certain forms of migraine that have been shown to activate the STN (*Schwedt et al., 2014*).

# Materials and methods

## Electrophysiology

### Animals

Fifty male Hooded Lister rats (265–450 g) were anaesthetised with an intra-peritoneal injection of urethane (ethyl carbonate; 1.25 g/kg as a 25% aqueous solution) and mounted in a stereotaxic frame with the skull level. Body temperature was maintained at 37°C with a thermostatically controlled heating blanket. Two stainless steel electrodes (E363-1, Plastics One, Roanoke, VA) were inserted into the left hindpaw, one under the skin of the plantar surface of the foot and the other under the skin of the medial aspect of the lower leg/ankle. Craniotomies were then performed to allow access to the STN and SC or PBN. In accordance with the policy of Lyon1 University and the Grenoble Institut des Neurosciences (GIN) and with French legislation, experiments were performed in compliance with the European Community Council Directive of November 24, 1986 (86/609/EEC). The research was authorized by the Direction Départementale des Services Vétérinaires de l'Isère – Ministère de l'Agriculture et de la Pêche, France (Coizet Véronique, PhD, permit number 381003). Every effort was made to minimize the number of animals used and their suffering during the experimental procedure. All procedures were reviewed and validated by the 'Comité éthique du GIN n°004' agreed by the research ministry (permits number 309 and 310).

## STN recordings

Extracellular single unit recordings were made from STN neurons located contralaterally to the stimulated hindpaw, using glass microelectrodes pulled via a vertical electrode puller (Narashige Laboratory Instruments Ltd. Tokyo, Japan) and broken back to a tip diameter of approximately 1 µm (impedances 5–20 MΩ, measured at 135 Hz in 0.9% NaCl). Electrodes were filled with 0.5 M saline and 2% Pontamine Sky Blue (BDH Chemicals Ltd., Poole, UK). The electrode was lowered into the STN (3.6–4.16 mm caudal to bregma, 2.0–3.0 mm lateral to midline, 6.8–8.20 mm ventral to the brain surface according to the atlas [*Paxinos and Watson, 2005*]) with a hydraulic microdrive (Trent Wells Inc.). The STN electrode was lowered until a putative STN neuron was identified on the basis of several criteria as follows. i) the pattern of activity while lowering the electrode, which was as follows: an initial absence of activity corresponding to the medial lemniscus fibre track, followed shortly after by large-amplitude, fast-bursty neurons located in zona incerta, and then a second absence of action potentials. The return of activity corresponded to the STN. ii) STN firing rate between 8.5 and 14.7 Hz (*Hassani et al., 1996*; *Kreiss et al., 1996*). iii) STN firing pattern described as irregular or bursting (*Hamani et al., 2004*).

## SC/PBN recording and muscimol experiments

Extracellular multiunit recordings were made simultaneously from the SC or PBN ipsilateral to the STN recording electrode using a tungsten electrode coupled to a 30-gauge stainless steel injector filled with muscimol (0.25 µg/µl in saline, Sigma-Aldrich). An angled approach was used in the PBN, with the electrode tilted caudally by 35°, entering the brain at 11.4 mm caudal to bregma and 1.9–2.0 mm lateral to midline. PBN was encountered 5.2–5.8 mm below the brain surface. In a second group of rats, the electrode/injector assembly was introduced vertically into the SC (AP: 6.2–6.5 mm, bregma; ML: 2.1–2.2 mm, bregma; DV: 4.2–4.5 mm, brain surface). The electrode/injector assembly was lowered into the SC into the lateral part of the deep layers of the SC, known to project to the STN (*Coizet et al., 2009*).

Microinjections were made (0.5 µl at a rate of 0.5 µl/min) via a 10 µl Hamilton syringe mounted on an infusion pump, connected to the injector by a length of plastic tubing. Extracellular voltage excursions were amplified, band-pass filtered (300 Hz–10 kHz), digitized at 10 kHz and recorded directly onto computer disc using a Micro 1401 data acquisition system (Cambridge Electronic Design [CED] Systems, Cambridge, UK) running CED data capture software (Spike 2).

## Stimulation procedure

As previously described (*Coizet et al., 2006*, *2010*), PBN and SC neurons were identified by their response to noxious footshocks induced by single pulses (0.5 Hz, 2 ms duration) at an intensity of 5.0 mA. The activity of the cells (single unit in STN and multiunit activity in the PBN or SC) was recorded during a control period (120 trials of sham stimulation) and during the application of noxious footshocks (120 trials). For the muscimol experiments, an injection of muscimol was made into the PBN or SC. Typically, a change in local SC/PBN multiunit activity was seen within 60–120 s of the injection. Noxious electrical footshock stimulation was applied throughout this period, until either the effects of the drug wore off in the SC/PBN, or the STN cell was lost. After a complete trial, further STN neurons were tested in the same way. Between 1 and 5 STN cells were tested in a single subject.

## Nociceptive nature of the stimulation

The electrical stimulation parameters from 3 to 5 mA have previously been shown to be approximately three times the threshold for activating C-fiber (*Chang and Shyu, 2001*; *Matthews and Dickenson, 2001*; *Carpenter et al., 2003*), to produce reliable Aδ and C-fiber responses in the anesthetized rat spinal cord (*Urch et al., 2003*) and to produce c-fos expression in the nociceptive superficial lamina of the spinal cord (*Coizet et al., 2006*). They also activate the SC (*Coizet et al., 2006*), the PBN (*Coizet et al., 2010*) and the dopaminergic neurons (*Coizet et al., 2006*; *2010*) in a qualitatively similar way as a mechanical noxious pinch with a teethed forceps.

To ascertain the noxious nature of our stimulation, we performed three control tests based on the previous observations. i) We first performed an intensity test on a group of STN cells following footshocks with intensities ranging from 0 to 5 mA. Among the 17 STN neurons responding to the

maximum 5 mA stimulation, a decrease of intensity was followed by a decrease of the number of responding cells following a 4 mA (n = 12) and a 3 mA (n = 7) stimulation. None of those cells showed a response for intensities under the threshold of 3 mA (*Figure 10A*, supplementary results). Furthermore, analysis performed on the 7 cells responding to 3, 4 and 5 mA showed a significant effect of the intensity of the stimulation when considering the maximum amplitude (ANOVA – repeated measure: F[2,20] = 11.51, p<0.01) and the magnitude (ANOVA – repeated measure: F[2,20] = 17.55, p<0.001). In both cases, there was a significant increase in the response parameters with the increase in the intensity (Tukey-Kramer post-hoc comparison — maximum amplitude: 3 mA vs 5 mA, p<0.01, 4mA vs 5 mA, p<0.05; magnitude: 3 mA vs 4 mA, p<0.05; 3 mA vs 5 mA, p<0.001). ii) We compared the effects of a mechanical pinch with a teethed forceps and a 5 mA foot-shock on STN neuron responses. All the cells responding to the manual pinch showed a qualitatively similar excitation to the footshock (*Figure 10B*, supplementary results), while none of the cells that were unresponsive to pinch were activated by footshock. iii) As previously reported (*Coizet et al., 2006*, *2010*), noxious footshock of 5mA for an hour induced the expression of c-fos labeling within the medial part of the ipsilateral superficial layers of the lumbar cord, especially layers I and II, although layers III–V also contained some labeling. In control animals, where electrodes were implanted but the stimulation was not delivered, substantially lower levels of c-fos were observed (*Figure 10A*). These results confirm that the footshock used in the present study was activating nociceptive elements in the lumbar spinal cord (*Besson and Chaouch, 1987*; *Almeida et al., 2004*) consistent with known somatotopic representations of the hindfoot; that is primary afferents from the foot terminate medially (*Swett and Woolf, 1985*).

## Histology and analysis

The position of SC and PBN recording sites were marked with a small lesion caused by passing 10 µA DC current for 2.5 min through the tungsten recording electrode. The final recording site for the STN recording electrode was marked by passing a constant cathodal current of 27.5 µA (constant current source) through the electrode for a period of 30 min to eject Pontamine Sky Blue. Animals were then killed by an overdose of pentobarbital and perfused with 0.9% saline followed by 4% paraformaldehyde. Brains were removed and postfixed overnight in 4% paraformaldehyde at 4°C, before being transferred into sucrose for 36 hr. Serial coronal (30 µm) sections were cut, mounted on slides and processed with a Nissl stain (Cresyl Violet). Once sections had been processed, recording sites were reconstructed onto sections taken from the atlas of *Paxinos and Watson (2005)*.

Peri-stimulus time interval histograms (PSTHs) were constructed based on SC/PBN multi-unit (bin width 1 ms) and STN single-unit data (bin width 10 ms). PSTHs were imported into an Excel program (Peter Furness, University of Sheffield; *Coizet et al., 2006*, *2009*, *2010*; *Dommett et al., 2005*), which determined the following response characteristics. (i) Baseline activity: the mean number of single multi-unit events during the 500 ms bins prior to the footshock. (ii) Response latency: the latency of a visually evoked response was marked as the point when the value of post-stimulation events exceeded 1.96 S.D. of the baseline mean. (iii) Response duration: response offset was recorded when post-stimulation activity returned to a value 1.96 S.D. of the baseline mean. iii) Amplitude of the response: the maximum amplitude during the response. (iv) Magnitude of the response: the mean number of single multi-unit events between response onset and offset minus the baseline mean.

## Lesion procedure in SC and PBN

Fourteen rats received a unilateral ibotenic acid lesion of the SC or the PBN. Each rat was anesthetized with isofluorane (5% for the induction and 1–2% for maintenance) and placed in a stereotaxic instrument. A 30-gauge metal injector needle filled with ibotenic acid (20 µg/µl in phosphate buffered saline) was introduced using the same coordinates as for the electrophysiological procedure. The injections in the PBN were made according to a previously published procedure by *Reilly and Trifunovic (2000*, *2001)* with electrophysiological guidance to improve the accuracy of the location of the lesion. The microinjections were made (0.5 µl/min) in the SC (0.5–0.65 µl) and the PBN (0.3–0.5 µl) as for the muscimol injections (see above). The cannula remained in situ for a further 10 min to minimize the spread of neurotoxin back along the track before the cannula was removed and the incision was closed.

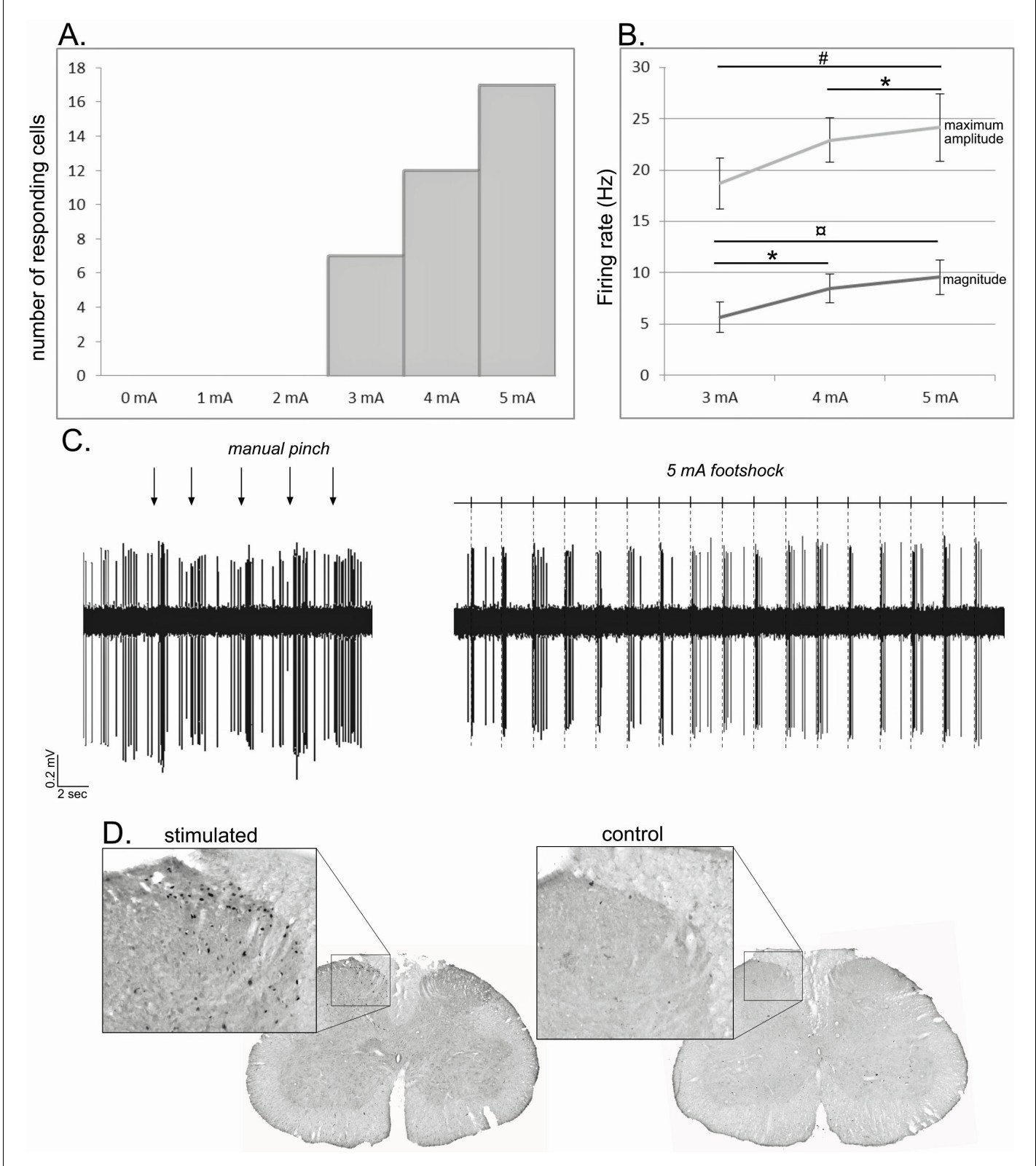

**Figure 10.** Noxious footshock. (A) Histogram showing the increase in the number of responding cells with the increase of the footshock intensity. (B) Increase of the maximum amplitude and magnitude of phasic nociceptive-evoked responses with the increase of the footshock intensity. (C) Individual example of an STN cell excited both by a mechanical noxious stimulation (pinch – left) and a 5 mA noxious footshock (right). (D) Coronal sections of the

*Figure 10 continued on next page*

*Figure 10 continued*
lumbar region of the spinal cord processed for c-fos expression in an animal subjected to 1 hr unilateral noxious electrical stimulation of the hindpaw (left) and in a control animal in which the electrodes were implanted into the hindpaw, but no footshock applied.
DOI: https://doi.org/10.7554/eLife.36607.014

## Lesion procedure in STN and hotplate test

Twenty Long Evans rats were anesthetized with ketamine (100 mg/Kg, s.c., Imalgène 1000, Merial, Lyon, France) and medetomidine (0.85 mg/Kg, s.c., Domitor, Orion Pharma, Espoo, Finland). Rats were secured in Kopf stereotaxic apparatus. Then, a unilateral 30-gauge stainless-steel injector needle connected by Tygon tubing (Saint Gobain performance plastics) with a 10 µL Hamilton microsyringe (Bonaduz, Switzerland) fixed on a micropump (CMA, Kista, Sweden) was positioned into the STN. Coordinates for the aimed site were (with tooth bar set at −3.3 mm): anteposterior −3.72 mm; lateral ±2.4 mm from bregma; dorsoventral −8.4 mm from skull (*Paxinos and Watson, 2005*). Rats received a bilateral injection of either ibotenic acid (9.4 µg/µL, AbCam Biochemical, Cambridge, UK; STN-lesioned group, n = 12) or vehicle solution (phosphate buffer, 0.1M; Sham control group, n = 8). The volume injected was 0.5 µL per side infused over 3 min. The injectors were left in place for 3 min to allow diffusion. At the end of surgery, medetomidine was reversed by 0.2 mg (4.28 mg/Kg, s.c.) atipamezole (Antisedan, Orion Pharma, Espoo, Finland). Three weeks after the surgery, all the animals were subjected to the hotplate test. Each rat was placed on a heated metal plate (53°) surrounded by a transparent cylinder. The experimenter was constantly watching the rat's behaviour during the test to measure the latency of the first sign of paw licking or jumping and to remove the animals from the apparatus quickly. The maximum time on the hot place was set to 30 s. The rat's behaviour was also video recorded online on the computer for a second finer analysis.

## 6-OHDA lesions

Rats were anesthetised with an intraperitoneal injection of a mixture of ketamine-xylazine (0.765/1.1 ml; 1 ml/kg, i.p.) and placed in a stereotaxic frame with the skull level. All the microinjections were made via a sharpened 30G injection cannula connected with polyethylene tubing to a 10 µl Hamilton syringe driven by an infusion pump (0.5 µl/min). After the injection, the cannula was left in place for a further 5 min to allow diffusion. Animals were divided into two groups: i) those with a total dopaminergic lesion (n = 9), in which 3 µl of 6-OHDA (Sigma-Aldrich, 3 mg/ml in sterile 0.9% NaCl and 0.1% ascorbic acid) was injected into the left SNc using the following stereotaxic coordinates: AP: +3.0 mm; ML:+2.1 mm and DV:+2.4 mm from interaural zero mm; and ii) a control group with no injection of the toxin (n = 9).

The extent of the DA denervation following the 6-OHDA injections in the SNc was determined using tyrosine hydroxylase (TH) immunohistochemistry. To reveal TH, the sections were washed and incubated in a blocking solution containing 0.1M PB with 0.3% of triton X-100 (TX), 2.5% of Bovine Serum Albumin (BSA) and 5% normal horse serum (NHS) for 2 hr before being transferred overnight to a 0.1M PB-TX 0.3% with 1% BSA and 2% NHS containing the primary mouse monoclonal TH antibody, diluted 1:3,000 (Chemicon, Hampshire, UK). The following day, sections were washed in 0.1M PB and incubated with the secondary antibody, biotinylated antimouse made in horse (in a dilution of 1:1,000 in 0.1M PB-TX 0.3% with 2% NHS) for 2 hr. Following further washes in 0.1M PB, the sections were exposed to the elite Vectastain ABC reagent (Vector Laboratories, Burlingame, CA, USA) diluted 1:100 in PB-TX 0.3%, for 2 hr. Again following washes in 0.1M PB, immunoreactivity was revealed by exposure to VIP (Vector Laboratories) for 2 min, which produced a purple reaction product. Sections were then mounted onto gelled slides, dehydrated through alcohols and cleared in xylene before being coverslipped with DPX. TH-immunolabelling of DA neurons and terminals was evaluated using a light microscope (Nikon, Eclipse 80i, TRIBVN, Chatillon, France) coupled to the ICS Framework computerized image analysis system (TRIBVN, 2.9.2 version, Chatillon, France). For quantification, TH-labeled coronal sections of SNc (AP −5.3 mm to −5.8 mm from Bregma) and striatum (AP 0.20 mm to −0.30 mm from bregma) were digitized using a Pike F-421C camera (ALLIED Vision Technologies Stradtroda, Germany). Optical densities (OD) were measured for the denervated and non-denervated territories of the lesioned animals for each section and were compared to those in the homologous regions of the sham-operated animals.

## Statistics

The statistical reliability of differences between response latencies for the SC/PBN and STN, and comparisons of response duration, amplitude and magnitude before and after SC/PBN injections of muscimol was made using parametric (ANOVA, t-test) or non-parametric (Wilcoxon, Mann-Whitney) statistical tests according to the normality of the data. STN baseline firing rate change before and after the noxious stimulation was assessed during the 500 ms before the sham and noxious stimulations. The data were imported in MATLAB, bined and compared using a Wilcoxon test. STN firing pattern was also assessed using MATLAB according to the methodology developed by *Piallat et al. (2011)*. Neurons were classified as irregular, regular or bursting according to the interspike distributions and autocorrelograms. Burst activity showed a wide or bimodal interval interspike distribution and a significant single peak on the autocorrelation function. Irregular activity was characterised by a wide interval interspike distribution and a flat autocorrelogram. Regular activity was characterised by a narrow interval interspike distribution and an autocorrelogram with multiple regular peaks.

### Anatomy
### Animals

We used 14 male Long Evans rats (350–460 g, Janvier, France). Animals were anaesthetised with an intraperitoneal injection of a mixture of Ketaset (0.765 ml/kg) and Rompun (1.1 ml/kg).

### Anterograde experiment

Single injections of the anterograde tracers biotinylated dextran amine (BDA: Sigma-Aldrich) or *Phaseolus vulgaris* leucoagglutinin (PHA-L: Vector Laboratories, Peterborough, UK) were made into the PBN. An angled approach was used as previously described (*Coizet et al., 2010*). BDA (10% in phosphate buffer; PB) was pressure ejected in volumes of 30–90 nl via a glass micropipette (20 μm diameter tip) using a compressed air injection system, while PHA-L was ejected iontophoretically (5μA anodal current applied to a 2.5% solution in PB, 7 s on/off for 15–20 min).

### Retrograde experiment

Small (10–20 nl) pressure injections of the retrograde tracers Cholera toxin subunit B (CTB, 1% solution in phosphate buffer) or the fluorescent tracer Fluorogold (FG, 4% in distilled water) were made into the STN. After allowing 7 days for the transport of tracers, animals were re-anesthetised with pentobarbitane and perfused transcardially. The brains were placed immediately in 4% PFA overnight before being cryoprotected by immersion in sucrose solution (20% in 0.1 M PB) for at least 36 hr. Three series of coronal or sagittal sections (30 μm) were cut on a freezing microtome and collected in 0.1 M PB for further immunohistochemistry processing, except for tissue containing FG where one series was collected directly onto slides, allowed to dry in a light protected box and coverslipped with DPX mountant.

### Histology

To reveal the tracers (BDA, PHA-L, CTB), free-floating sections were washed with 0.1 M PB followed by 0.1 M PB containing 0.3% Triton X-100 (PB-TX) for 30 min. For animals injected with PHA-L or CTB, the sections were incubated overnight in primary antibody solution (goat anti-PHA-L, 1:1,000 dilution, Vector, or goat anti-CTB, 1:4,000 dilution, Quadratech). The next day, sections were washed with PB-TX and incubated for 2 hr in biotinylated rabbit anti-goat IgG (1:100, Vector, in PB-TX containing 2% normal rabbit serum). After 30 min washing, all the sections were incubated in Elite Vectastain ABC reagent (Vector, 1:100 in PB-TX) for 2 hr. The peroxidase associated with the tracers was revealed by reacting tissue with $H_2O_2$ for approximately 1 min using nickel-enhanced diaminobenzidine (DAB) as the chromogen for BDA and CTB (black reaction product) and using nickel-free DAB for PHA-L (brown reaction product). Finally, sections were washed in PB, mounted on gelatine-coated slides, dehydrated in graded dilutions of alcohol, cleared in xylene and coverslipped in DPX.

### Analysis

Following injections of anterograde tracers into the PBN, three coronal sections through the STN separated by ~0.5 mm (equivalent to −3.6, −3.8 and −4.1 mm caudal to bregma in the atlas of

*Paxinos and Watson (2005)* or three sagittal sections (equivalent to 1.9, 2.4 and 2.9 lateral to bregma) were selected for analysis. Sections of interest were digitized using a light microscope (Nikon, Eclipse 80i, TRIBVN, 2.9.2 version, Chatillon, France) coupled to the ICS Framework computerized image analysis system (TRIBVN, 2.9.2 version, Chatillon, France) and a Pike F-421C camera (ALLIED Vision Technologies Stradtroda, Germany).

The location of retrogradely labelled cells was plotted on four coronal sections through the PBN separated by ~0.5 mm (equivalent to −8.8, −9.3, −9.8 caudal to bregma). A series of digital images (magnification 10 X) were taken and imported into a graphics program (Macromedia Freehand) where they were montaged. The borders and layers of the PBN were drawn over the montage. The location of anterogradely labelled axons and terminals was plotted on sagittal sections from 1.9 to 2.9 lateral from bregma. Fibres and terminals associated with the injections were traced with the aid of a pen tablet (intuos) with a microscope equipped with a camera lucida and a graphics program (Microsoft PowerPoint). By focusing on different depths in the brain sections, it was possible to produce a drawing that contained all labelled elements in the section.

For the 3D reconstruction, the most representative example was selected and series of sagittal sections containing PBN and STN were digitized using a light microscope (Nikon, Eclipse 80i, TRIBVN, 2.9.2 version, Chatillon, France) coupled to the ICS Framework computerized image analysis system (TRIBVN, 2.9.2 version, Chatillon, France) and a Pike F-421C camera (ALLIED Vision Technologies Stradtroda, Germany). Digitized images were converted into a .tiff format and individually exported into Adobe Photoshop to create individual .tiff files with the same dimensions. A stack of 2D sagittal sections was then created using Cygwin (Cygwin TM sources) and IMOD package software (Boulder laboratory for 3D Electron Microscopy of Cells, University of Colorado, Boulder, CO) (*Kremer et al., 1996*). As previously described (*Coizet et al., 2017*; *Mailly et al., 2010*), sections were aligned with the Midas program from IMOD using manual rigid body transformations. The stack was opened in IMOD, where the structures of interest were delineated including the PBN, SNr/SNc, STN, and SC. The injection site and individual ascending labelled axons were drawn in IMOD directly on the digitized images. This process also created a 3D reconstruction of ascending fibers.

## Acknowledgements

The Photonic Imaging Center of Grenoble Institute Neuroscience (Université Grenoble Alpes – Inserm U1216) is part of the ISdV core facility and certified by the IBiSA label. Région Rhône-Alpes.

## Additional information

### Funding

| Funder | Author |
| --- | --- |
| Institut National de la Santé et de la Recherche Médicale | Veronique Coizet |
| ADR Région Rhône Alpes | Veronique Coizet |
| UGA AGIR-POLE | Veronique Coizet |

The funders had no role in study design, data collection and interpretation, or the decision to submit the work for publication.

### Author contributions

Arnaud Pautrat, Marta Rolland, Margaux Barthelemy, Formal analysis, Investigation; Christelle Baunez, Investigation, Methodology, Writing—review and editing; Valérie Sinniger, Investigation, Methodology; Brigitte Piallat, Software, Formal analysis, Methodology; Marc Savasta, Funding acquisition, Writing—review and editing; Paul G Overton, Conceptualization, Project administration, Writing—review and editing; Olivier David, Formal analysis, Funding acquisition, Methodology, Writing—review and editing; Veronique Coizet, Conceptualization, Formal analysis, Supervision, Funding acquisition, Validation, Methodology, Writing—original draft, Project administration, Writing—review and editing

## Author ORCIDs
Christelle Baunez (iD) https://orcid.org/0000-0002-4368-652X
Veronique Coizet (iD) http://orcid.org/0000-0001-5192-6610

## Ethics

Animal experimentation: In accordance with the policy of Lyon1 University, the Grenoble Institut des Neurosciences (GIN) and the French legislation, experiments were done in compliance with the European Community Council Directive of November 24, 1986 (86/609/EEC). The research was authorized by the Direction Départementale des Services Vétérinaires de l'Isère - Ministère de l'Agriculture et de la Pêche, France (Coizet Véronique, PhD, permit number 381003). Every effort was made to minimize the number of animals used and their suffering during the experimental procedure. All procedures were reviewed and validated by the "Comité éthique du GIN no 004" agreed by the research ministry (permits number 309 and 310).

## Decision letter and Author response
Decision letter https://doi.org/10.7554/eLife.36607.017
Author response https://doi.org/10.7554/eLife.36607.018

# Additional files

## Supplementary files
• Transparent reporting form
DOI: https://doi.org/10.7554/eLife.36607.015

## Data availability
All data generated or analysed during this study are included in the manuscript and supporting files. Matlab scripts used to analyse the data are freely available on the ImaGIN platform website (https://f-tract.eu/software/imagin/ ; copy of the source code archived at https://github.com/elifesciences-publications/ImaGIN2_source_code).

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
