## [Decision Letter]

[Editors’ note: this article was originally rejected after discussions between the reviewers, but the authors were invited to resubmit after an appeal against the decision.]

Thank you for submitting your work entitled "Revealing a novel nociceptive network that links the subthalamic nucleus to pain processing" for consideration by *eLife*. Your article has been reviewed by three peer reviewers, including Peggy Mason as the Reviewing Editor and Reviewer #1, and a Senior Editor.

Our decision has been reached after consultation between the reviewers. Based on these discussions and the individual reviews below, we regret to inform you that your work will not be considered further for publication in *eLife*.

The reviewers saw this study as well motivated into a largely un-explored aspect of PD and DBS therapy. Yet the reviewers had several serious reservations about the data:

- The experiments were underpowered. Results for n-s of 1 or 2 were discussed and many to most of the component experiments had total n-s of under 6.

- There are 3 different lesion methods used. The muscimol and ibotenic acid are used on cell responses and give very different results. An electrolytic lesion is used on behavior. Why? and if the neuronal responses underlie the behavior why the change in inactivation method?

- The absence of placement controls, and thus no control for repeated testing, for the lesions is a concern.

- The relationship between the tonic and phasic responding cells was unclear. From the numbers it would appear that some cells had both types of responses. Furthermore, regarding the tonic responses, some discussion of whether these were responses to noxious stimulation or to arousal would be helpful.

*Reviewer #1:*

This is a well motivated study into an understudied symptom of PD, namely spontaneous pain, that appears relieved by STN-DBS. This study looks at mechanisms that may underlie this DBS effect. The authors find nociceptive responses in STN, that appear to arise in PBN and are positively modulated by Superior Colliculus. Electrolytic lesions of STN attenuate the hot plate test, lengthening the latency to respond. There are projections from the SC and PBN to STN.

This is a thorough paper. My major issue is that the authors use three different lesioning methods – electrolytic, muscimol and ibotenic acid- none of which are specific to cell type. Why the different lesioning methods yield different results is unclear. Thus the different roles of SC and PBN in shaping the nociceptive responses of STN remain unclear.

I don't understand Figure 2B. On left is supposed to be a decrease but if the dotted line is stim onset as mentioned in legend, then the rasters show an increase and vice versa for the right panel. Furthermore what is the relationship between the phasic and tonic responses?% s would suggest that they have to overlap. The statement that "Interestingly, the orientation of the change could be predicted by the spontaneous firing rate of STN cells…" is an overstatement. Only a decrease is associated with a higher rate. A lower rate is not informative as to whether the cell's firing rate will increase or not change.

The absence of placement controls for the lesions is a concern. There is simply no control and the effect of repeated testing is not known. Furthermore, this would appear to be a very important piece of the experiment and the lack of an accompanying figure is curious.

*Reviewer #2:*

In this manuscript, Arnaud Pautrat and colleagues show that the subthalamic nucleus (STN) is linked to the nociceptive brain network, involved in pain processing, affected by dopamine depletion and therefore probably play a major role in pathophysiology of the pain symptoms of Parkinson's disease.

They do so by revealing that 1) rodent (under urethane anesthesia) STN neurons exhibit in-vivo complex tonic and phasic responses to noxious stimuli; 2) pain perception is altered following a lesion of STN. They further characterize physiologically (by muscimol injections and simultaneous recording in the STN and at the inactivated target, and by ibotenic acid lesions) the role of the superior colliculus (SC) and the parabrachial nucleus (PBN) in the transmission of nociceptive information to the STN (PBN inactivation was found to be more effective than SC inactivation at reducing STN phasic pain responses). Additionally, they provide anterograde and retrograde anatomical evidence for the connections between the parabrachial nucleus and the STN (the anatomical connections between the SC and the STN are well known). The authors thus conclude that the PBN acts as a critical source of nociceptive input to the STN and the SC as a critical modulator of those responses. Finally, they show that STN pain responses are abnormal in the 6-OHDA rat model of PD.

This is a very comprehensive and important study. Unluckily, this reviewer is missing the answer to the next obvious question: do STN DBS affect pain processing in their model of Parkinsonism. If the authors have not done this experiment, please discuss this issue. My second major question is related to the discrimination between pain and arousal effects. I am aware of many rodent studies (e.g. by the Oxford group) that use pain to modulate the arousal (as reflected in the EEG) of rodents under urethane anesthesia. These points should be further discussed by the authors.

*Reviewer #3:*

This study examined electrophysiological responses to noxious stimuli in the rat STN in healthy rats and in a PD model, the role of the superior colliculus and parabrachial nucleus, and the effect of STN lesions on pain-related behaviour. Although it has been known for some time that the STN is involved in pain, STN DBS can relieve PD, and that patients with PD can have chronic pain, there is little understanding of the underlying mechanisms and so this study topic is important. However, the study consists of so many subparts (mostly underpowered) that are poorly presented in terms of rationale and outcomes, that making sense of the study is challenging.

- There are no hypotheses laid out and so this complex study feels like a series of experiments that are not clearly tied together with any clear framework or proposed mechanism being tested.

- I found the study very difficult to follow. Given that the study is quite complex with many experiments, approaches and manipulations, it was somewhat difficult to follow without a clear framework as noted above. Thus, it would be useful to have a figure that shows the main concept of the study (perhaps with an anatomical circuit diagram) and the approach with these sub-studies/experiments and why the manipulations were being done.

- The authors should be more careful with terminology such as "painful stimulation" (should use "noxious stimuli" instead), "pain-activated cells" (should use "noxious stimulus activated cells"), and "pain perception" as pertains to behaviours in the rat as there is no way of knowing if the animals are indeed experiencing pain.

- The study was restricted to male rats. This is a major flaw given that there is no reason to omit female rats from such a study and given that there may indeed be sex differences in the pain system, it is a surprising study design flaw. Some countries and granting agencies now require animal studies to be done in both sexes unless there is a clear justification for restricting the study to one sex.

- In the electrophysiological study in healthy rats, it is not clear what kind of cells are being recorded from in the STN. Were the cells tested for responsiveness to anything other than footshock? It would be important to know if they were responsive to motor manipulations.

- The STN lesion studies were underpowered and the data were not provided in detail in figures.

- The effects of other manipulations and the anatomical studies were poorly conveyed in the results and figures and as with the STN lesion studies, many were underpowered.

- The symbols in Figure 1 could be improved to distinguish more clearly the responsive from non-responsive cells.

- I may have missed it but please clarify if the footshock responses were ipsi or contralateral. Also, it is important to provide information as to how the parameters of the noxious stimulation was derived and known to be noxious in nature.

[Editors’ note: what now follows is the decision letter after the authors submitted for further consideration.]

Thank you for submitting your article "Revealing a novel nociceptive network that links the subthalamic nucleus to pain processing" for consideration by *eLife*. Your article has been reviewed by Peggy Mason as Reviewing Editor and Andrew King as the Senior Editor.

This revision incorporates a far better motivated Introduction that convinces the reader that the work is of importance for understanding an important symptom in PD and that nociceptive responses in STN may represent an important interrupt-signal. Figure 1 and the Introduction also improve the reader's understanding of what experiments were done and why.

Yet there remain many concerns with this manuscript. A large proportion of these concerns could go away with thorough copy-editing for common English usage and we recommend that you ensure that this to be done prior to any resubmission. Major concerns are:

It is not clear how the results inform DBS. The authors argue for DBS resolving PD-associated pain and they may be right. However, caution should be exercised in making this conclusion given that the STN responses to noxious stimulation were augmented in the rodent PD model. One possibility is that DBS stimulation exacerbates the pain symptoms of PD by exciting cells that are already pathologically facilitated. A second possibility is that PD stimulation does not excite STN cells (given that DBS stimulation is known to have effects commensurate with inhibition of DBS cells). The authors should at the very least discuss the direction of the DBS stimulation on net STN nociceptive responses.

As mentioned in the previous review, the authors should be careful regarding statements about pain vs. noxious stimulation. The papers quoted to support "pain can modulate background activity" involve the responses to somatic stimulation, *not* the presence of a pain percept. Attention to the pain – nociceptive distinction should also be applied to the whole section starting with "Therefore, the main objective of the present work…"

(other examples in subsection “Nociceptive responses in afferent structures: SC and PBN”, subsection “STN nociceptive responses in a rat model of Parkinson’s disease”, last paragraph, Discussion, first and fourth paragraphs, and many more places).

"we could differentiate three types of STN neurons, which either showed an increase, a decrease or no change of their baseline firing rate with the introduction of the noxious stimulation. The spontaneous firing rate of STN "down" cells was significantly higher compared to the two other groups, suggesting the possibility of a separate group of cells." I have two comments about this. First, the three "types" identified refer to the three possibilities. This is hardly a surprising finding unless the three types of responses are correlated with an independent metric. My second comment is that another possibility regarding the higher FR in down cells is that other cells may also be inhibited when their baseline FR increases which thereby allows a decrease to be detected.

---

## [Author Response]

[Editors’ note: the author responses to the first round of peer review follow.]

Two additional experiments have been added to fully explore the implications of our findings. This manuscript now includes in vivo electrophysiology, behavioral assessment, tract-tracing neuroanatomy and a rat model of Parkinson’s disease to characterize a new pain network involving the subthalamic nucleus. We think this work provides conceptual advances and is of general interest because: – Clinical work with Deep Brain Stimulation (DBS) clearly links together pain and the subthalamic nucleus, but neither the neural basis for that link, nor the mechanism of action of DBS – a widely used clinical technique – are known. We provide significant insights into both. – We show for the first time that the subthalamic nucleus exhibits complex mono- or multi-phasic responses to noxious stimuli as well as various changes in baseline firing, revealing previously uncharacterized separate subpopulations of cells in this structure. – Analysis of STN tonic pain responses reveals that STN cells are not uniformly computing this painful signal. Therefore, our data provide insights about the complex intrinsic organization of the STN to differentially modulate action selection. – We also show for the first time the afferent origin of the nociceptive signals to the subthalamic nucleus and how they may be modulated. Our results reveal that the subthalamic nucleus is linked to a nociceptive network, within which the effects of DBS can be understood. – We demonstrate that the subthalamic pain responses and link to a nociceptive network translate into a functional role for STN in mediating pain perception. We show that rat pain perception is altered following a lesion of the subthalamic nucleus. – Discovering the source of afferent sensory (pain) information to the subthalamic nucleus reveals another “hyper-direct” input pathway to the basal ganglia complementing that already known from the cerebral cortex. Complementing our previous work, it supports the strong anatomical and functional connections between the basal ganglia and sensori-motor structures from the brainstem. – On the clinical side, our demonstration that subthalamic neurons are involved in pain processing and its well-known dysfunction in Parkinson’s disease could explain some of the debilitating pain symptoms that are prevalent in this disease. Our data support this hypothesis with a clear demonstration that STN pain processing is altered in a rat model of this disease. Given that pain phenomena are a feature of many neurodegenerative disorders, including those involving the basal ganglia, our insights could have widespread implications for the pathogenesis of aberrant pain phenomena and for their treatment.

The reviewers saw this study as well motivated into a largely un-explored aspect of PD and DBS therapy. Yet the reviewers had several serious reservations about the data:- The experiments were underpowered. Results for n-s of 1 or 2 were discussed and many to most of the component experiments had total n-s of under 6.

In the majority of cases, the very small Ns arose from considering subdivisions within such things as the response types in the STN following noxious stimulation. Given that every response is in some way unique, the decision about where to demarcate the boundaries between response types is somewhat arbitrary and on reflection we may have been led to focus on smaller differences between responses when the major differences (and larger Ns) were actually the most interesting. The smaller categories of response can be subsumed quite easily. We have now changed the section of the manuscript dealing with the categorization of responses. This section now focuses exclusively on the higher order classes of response and subsequently retains ample sample sizes in all categories.

Regarding tract-tracing experiments, the question of small Ns comes up frequently. We and others find that in general, there is low inter-case variability for anatomical tract-tracing, and, as a result, typically just a few cases are needed to describe a specific projection (i.e. Berendse and Groenewegen, Neuroscience, 1991; Coizet et al.,; 2017; Comoli et al., Nature Neuroscience, 2003; Kita and Kitai, J of comparative neurology, 1987). All our injections of anterograde tracer in the parabrachial nucleus (n = 8) labeled terminals in the dorsal part of the STN. This finding is consistent with the classical definition of a dorsal sensory-motor compartment of the STN where most sensory-motor structures project (Hamani et al., 2004; Haynes and Haber, J Neuroscience, 2013; Coizet et al., 2009; Kita and Kitai, J of comparative neurology, 1987; Kolomiet et al., 2001; Lanciego et al., Eur J of Neuroscience, 2004). The strength of our data is further supported by the use of different anterograde tracers (PHAL and Dextran) and retrograde tracers (Fluorogold and CTB), which we show provide the same distribution of labelled cells and terminals. To re-assure the reviewers, additional CTB injections in STN, recently processed in our lab (and confirming the present data), can be added to make the Ns equal to those using Fluorogold. Plotting all our anatomical data would be excessive, which is why we have presented the two most representatives cases for each anterograde and retrograde experiment. This technique of presentation is also based on previous works including our own (i.e. Comoli et al., Nature Neuroscience, 2003; Coizet et al., J of comparative neurology, 2007; J of neuroscience, 2009, 2017).

- There are 3 different lesion methods used. The muscimol and ibotenic acid are used on cell responses and give very different results. An electrolytic lesion is used on behavior. Why? and if the neuronal responses underlie the behavior why the change in inactivation method?

We apologize that this was not sufficiently well explained. We used muscimol and ibotenic acidbecause each have strengths and weaknesses, but together they give a comprehensive picture of the involvement of the SC and PBN in the transmission of nociceptive information to the STN.

Muscimol does not create a lesion but produces a temporary pharmacological inactivation of the injected area (superior colliculus and parabrachial nucleus) allowing a direct comparison of STN nociceptive responses between a pre injection control period, a post injection period and a recovery period once the muscimol effect was cleared.

However, although the technique allows a pre-post comparison, the extent to which muscimol spreads within the target structures is difficult to determine. Hence, we also used ibotenic acid lesions whose extent could be determined histologically.

Those lesion/inactivation experiments gave the same results:

Pharmacological inactivation of the superior colliculus did not suppress STN noxious responses but rather diminished the duration of this responses (n = 9). Ibotenic acid lesion of the superior colliculus did not suppress STN noxious responses but rather diminished the duration of the responses (n = 28).

Pharmacological inactivation of the parabrachial nucleus suppressed or strongly reduced STN noxious responses (n = 13). Ibotenic acid lesion of the parabrachial nucleus strongly reduced STN noxious responses (n = 30).

NB: We did not use electrolytic lesion for the behavior but used ibotenic acid lesions, same as for the electrophysiological experiment.

The word “temporary” was added to the introductory paragraph of the Results “Effect of SC or PBN inhibition on STN nociceptive responses” for clarity: “To test the possibility that the SC or PBN transmits nociceptive signals to the STN, we pharmacologically inhibited their neuronal activity with muscimol, a GABAA agonist, and evaluated the effect of their temporary inactivation on STN nociceptive responses.”

- The absence of placement controls, and thus no control for repeated testing, for the lesions is a concern.

The hot place test was performed on two different groups of rat, a group with a ibotenic acid lesions of the STN (lesion group) and a group of sham rats injected in the STN with the excipient. Therefore, there was no repeat of the rats in this test.

- The relationship between the tonic and phasic responding cells was unclear. From the numbers it would appear that some cells had both types of responses. Furthermore, regarding the tonic responses, some discussion of whether these were responses to noxious stimulation or to arousal would be helpful.

This is an important question. The analysis of the tonic firing rate was performed on both STN nociceptive and non-nociceptive cells (n=98). This information has been added to the manuscript for clarity:

“We therefore performed an individual analysis on each of the 98 STN cells, whether responding to the noxious stimulation or not, to test if the change of their baseline firing rate after the introduction of the stimulation was statistically different or not robust …”.

There is no necessary relationship indeed between tonic and phasic responses as the modulation of tonic firing rate was also found in both nociceptive and non-nociceptive STN cells, although the reviewer is correct – both phasic and tonic modulations did occur in some (but not all) nociceptive cells. This information has been added to the manuscript for clarity:

“Contingency analysis did not reveal a specific topography of their location within STN, or link to their action potential shape, or to the presence of a phasic response or not.”

In addition to the above, please find below our point-by-point response addressing all the reviewer’s remaining comments:

Reviewer #1:

[…] This is a thorough paper. My major issue is that the authors use three different lesioning methods – electrolytic, muscimol and ibotenic acid- none of which are specific to cell type. Why the different lesioning methods yield different results is unclear. Thus the different roles of SC and PBN in shaping the nociceptive responses of STN remain unclear.I don't understand Figure 2B. On left is supposed to be a decrease but if the dotted line is stim onset as mentioned in legend, then the rasters show an increase and vice versa for the right panel. Furthermore what is the relationship between the phasic and tonic responses?% s would suggest that they have to overlap. The statement that "Interestingly, the orientation of the change could be predicted by the spontaneous firing rate of STN cells…" is an overstatement. Only a decrease is associated with a higher rate. A lower rate is not informative as to whether the cell's firing rate will increase or not change.

Figure 2B has been changed (now Figure 3 in the manuscript). We have removed the confusing phasic response occurring with the stimulation itself (at 0), which is not of interest for this analysis. Baseline firing rate is measured during the 500 ms (-0.5 to 0 in the figure) before the stimulation (at 0). We compared this value between the control set of stimulations (stimulator turned off) and the noxious set of stimulations (stimulator turned on). These two individual examples of an increase and decrease of baseline firing rate have been added for clarity. We wanted to illustrate our point that the baseline firing rate (the ‘tonic response’) of the cells was differentially changed when the animals were subjected to noxious stimulation, in some cases the baseline was increased and in other cases it was reduced.

The absence of placement controls for the lesions is a concern. There is simply no control and the effect of repeated testing is not known. Furthermore, this would appear to be a very important piece of the experiment and the lack of an accompanying figure is curious.

We added a figure to complement the results (Figure 4 in the manuscript).

Reviewer #2:

[…] This is a very comprehensive and important study. Unluckily, this reviewer is missing the answer to the next obvious question: do STN DBS affect pain processing in their model of Parkinsonism. If the authors have not done this experiment, please discuss this issue.

Testing the effect of STN DBS on pain processing in our rat model require a full set of experiments aiming to answer the following questions, partly raised by the present data:

1) Does STN DBS affect pain processing in our rat model of Parkinson’s disease, as suggested.

2) What would be the best target within STN? Pain is a multifaceted experience that can be understood in terms of somatosensory, affective and cognitive dimension. DBS therapies focused on a single facet of pain, originally targeting either somatosensory networks or more recently targeting affective regions (Shirvalkar et al., 2018). The STN is a small structure with functional territories such as limbic, cognitive and sensory, in close proximity to each other. This would allow the potential modulation of different modalities of pain. The best DBS electrodes placement within those territories would have to be tested.

3) What would be the best parameters of stimulation to re-establish normal pain processing? STN DBS parameters have been set to improve the motor symptoms. However, numerous nonmotor symptoms can worsen or improve depending on the electrical stimulation parameters, as well as the location of the electrode (i.e. Kim et al., 2015). This would have to be tested with respect to pain symptoms.

4) As the STN appears to be part of a novel nociceptive network, how are aspects of that network modulated to achieve a DBS-related analgesic effect, should one be achieved?

As expressed by reviewers 1 and 3, this is already a dense article, and the experiments planned to answer that question would constitute a second paper. The issue and the complexities thereof can however be discussed more fully in the present manuscript.

A paragraph has been added in the Discussion: “Further experiments are now needed to fully characterize the effect of STN-DBS on pain processing in our rat models and how aspects of that network are modulated to achieve a DBS-related analgesic effect. […] The best parameters of stimulation for pain would need to take into account the effect of those parameters on other symptoms of PD.”

My second major question is related to the discrimination between pain and arousal effects. I am aware of many rodent studies (e.g. by the Oxford group) that use pain to modulate the arousal (as reflected in the EEG) of rodents under urethane anesthesia. These points should be further discussed by the authors.

This is an important point. Pinching the rat’s hindpaw for a few seconds is indeed a technique used to generate prolonged period of cortical activation in anesthetized rats (i.e. Mallet et al., 2008; Sharott et al., 2017). We therefore expect to generate this cortical activation during our stimulation protocol. We think that cortical activation is necessary for the expression of STN nociceptive responses, and also for the flow of low-level sensory information coming from brainstem structures such as the superior colliculus and the parabrachial nucleus. Indeed, our hypothesis is that the STN represents a node by which the cortex can control the activity of brainstem structures. Under normal circumstances, the STN receives numerous sensory inputs such as visual (Coizet et al., 2009) and auditory (Pautrat et al., in preparation) from the superior colliculus, and pain from the parabrachial nucleus (present data). With STN involvement of the brain’s interrupt circuitry, set to terminate the current behavior to allow the selection of a more appropriate action, such a circuit would require oversight to make sure behaviours were not constantly interrupted. Our suggestion is that brainstem inputs to the STN require a control signal from the cortex to favor or ignore those sensory processing. We therefore hypothesize that an increase arousal would facilitate STN responses to noxious stimulation to terminate behavior that achieves a painful outcome. It would be interesting in the future to test whether the noxious responses described in our current work vary with cortical activity.

Reviewer #3:

This study examined electrophysiological responses to noxious stimuli in the rat STN in healthy rats and in a PD model, the role of the superior colliculus and parabrachial nucleus, and the effect of STN lesions on pain-related behaviour. Although it has been known for some time that the STN is involved in pain, STN DBS can relieve PD, and that patients with PD can have chronic pain, there is little understanding of the underlying mechanisms and so this study topic is important. However, the study consists of so many subparts (mostly underpowered) that are poorly presented in terms of rationale and outcomes, that making sense of the study is challenging.- There are no hypotheses laid out and so this complex study feels like a series of experiments that are not clearly tied together with any clear framework or proposed mechanism being tested.

We apologize that this was not sufficiently explained. The main objective of the study was to establish and characterize the link between pain and the STN. For that purpose, we performed series of experiment to answer series of questions related to this main objective:

- We had to demonstrate that STN was processing pain information.

- We also had to demonstrate that manipulation of the integrity of STN could change physiological pain responses.

- In addition to these physiological results, it was crucial to demonstrate that the STN was anatomically targeted by afferents originating from primary pain structures. Indeed, there is to date no clear description of a link between the basal ganglia and the cerebral network involved in pain.

- Finally, to support the hypothesis that STN dysfunction could underlie some pain symptoms observed in Parkinson’s disease, we had to show that the STN was indeed dysfunctional in a rat model of this disease.

We think we achieved to answer to all those questions.

To clarify our objectives, we rephrased the last paragraph of the Introduction:

“Therefore, the main objective of the present work was to characterize the link between STN and pain, answering the following questions:

- Does the STN process pain information? We explored the possibility that noxious stimulation would induce nociceptive responses in the rat STN in vivo with electrophysiology. […] We present convergent evidence that the STN is functionally linked to a nociceptive network and that its pain responses are affected in parkinsonism. The objectives above are summarized in Figure 1.”

- I found the study very difficult to follow. Given that the study is quite complex with many experiments, approaches and manipulations, it was somewhat difficult to follow without a clear framework as noted above. Thus, it would be useful to have a figure that shows the main concept of the study (perhaps with an anatomical circuit diagram) and the approach with these sub-studies/experiments and why the manipulations were being done.

A figure summarizing the objectives and methods has been added to the manuscript (Figure 1).

- The authors should be more careful with terminology such as "painful stimulation" (should use "noxious stimuli" instead), "pain-activated cells" (should use "noxious stimulus activated cells"), and "pain perception" as pertains to behaviours in the rat as there is no way of knowing if the animals are indeed experiencing pain.

We changed “painful stimulation” and “pain-activated cells” by “noxious stimuli” and “noxious stimulus activated cells” as suggested. “Pain perception” has been replaced by “pain responses”.

- The study was restricted to male rats. This is a major flaw given that there is no reason to omit female rats from such a study and given that there may indeed be sex differences in the pain system, it is a surprising study design flaw. Some countries and granting agencies now require animal studies to be done in both sexes unless there is a clear justification for restricting the study to one sex.

This is an important point. We could justify the use of male rats by the fact that our present data rely on experiments performed previously in our lab (and elsewhere) on male rats. Our original characterization of the tecto-subthalamic pathway and electrophysiological experiments using visual stimulation were conducted using male rats (Coizet et al., 2009). This experiment set up the hypothesis that the superior colliculus, processing nociceptive information, could therefore transmit this information to the subthalamic nucleus. Furthermore, the characterization of the parabrachial-nigral pathway and electrophysiological experiments using noxious stimulation was also on male rats (Coizet et al., 2006, 2010). This work set up the hypothesis that the parabrachial nucleus could also be a candidate to relay nociceptive information to the subthalamic nucleus.

The fact that we used male rats only has been highlighted in the discussion: “However, there is a note of caution required since we used only male rats, and thus care should be taken in extrapolating our results to females.”

- In the electrophysiological study in healthy rats, it is not clear what kind of cells are being recorded from in the STN. Were the cells tested for responsiveness to anything other than footshock? It would be important to know if they were responsive to motor manipulations.

A group of 24 noxious stimulus activated cells were tested for non-noxious somato-sensory responses. As indicated in the results, only three cells exhibited an excitation to somatosensory stimulation therefore the majority of the cells recorded were nociceptive cells only. STN cell motor responses were not tested.

- The STN lesion studies were underpowered and the data were not provided in detail in figures.

We performed a power analysis on the groups tested in the hot plate test:

t tests – Means: Difference between two independent means (two groups)

Analysis: Post hoc: Compute achieved power

Input: Tail(s) = Two

Effect size d = 1.2816962

α err prob = 0.05

Sample size group 1 = 8

Sample size group 2 = 12

Output: Noncentrality parameter δ = 2.8080557

Critical t = 2.1009220

Df = 18

Power (1-β err prob) = 0.7567259

The result of this analysis confirmed the correct number of animals for this experiment. A figure of the lesion in STN has been added.

- The effects of other manipulations and the anatomical studies were poorly conveyed in the results and figures and as with the STN lesion studies, many were underpowered.

This point has been answered in the first part of the main points raised by the reviewers.

- The symbols in Figure 1 could be improved to distinguish more clearly the responsive from non-responsive cells.

Changed.

- I may have missed it but please clarify if the footshock responses were ipsi or contralateral. Also, it is important to provide information as to how the parameters of the noxious stimulation was derived and known to be noxious in nature.

The footshock responses were contralateral. For clarity, we added this information into the Results section “Phasic response: Following the noxious stimulation performed in the contralateral hindpaw (120, 0.5 Hz), 19 STN cells remained unresponsive (19% ) while 79 STN neurons (81% ) exhibited a phasic response to the footshock with several patterns of response.”

An early draft included a paragraph explaining our choice of parameters based on the data from the literature and an additional pre-experiments performed to ascertain the noxious nature of our stimulations:

“Nociceptive nature of the stimulation

The electrical stimulation parameters from 3 to 5 mA have previously been shown to be approximately three times the threshold for activating C-fiber (Chang and Shyu, 2001; Matthews and Dickenson, 2001; Carpenter et al., 2003), to produce reliable Aδ and C-fiber responses in the anesthetized rat spinal cord (Urch et al., 2003) and c-fos expression in the nociceptive superficial lamina of the spinal cord (Coizet et al., 2006). […] These results confirm that the footshock used in the present study was activating nociceptive elements in the lumbar spinal cord (Besson, 1987; Almeida et al., 2004) consistent with known somatotopic representations of the hindfoot; i.e. primary afferents from the foot terminate medially (Sweet and Woolf, 1985)”.

This paragraph has been adapted and included in the Materials and methodsand Figure 10 included in the supplementary results.

[Editors’ note: the author responses to the re-review follow.]

[…] Yet there remain many concerns with this manuscript. A large proportion of these concerns could go away with thorough copy-editing for common English usage and we recommend that you ensure that this to be done prior to any resubmission. Major concerns are:It is not clear how the results inform DBS. The authors argue for DBS resolving PD-associated pain and they may be right. However, caution should be exercised in making this conclusion given that the STN responses to noxious stimulation were augmented in the rodent PD model. One possibility is that DBS stimulation exacerbates the pain symptoms of PD by exciting cells that are already pathologically facilitated. A second possibility is that PD stimulation does not excite STN cells (given that DBS stimulation is known to have effects commensurate with inhibition of DBS cells). The authors should at the very least discuss the direction of the DBS stimulation on net STN nociceptive responses.

The local effect of STN DBS on STN neurons has been added in the Discussion as follows:

“DBS effects are complex and despite the success of DBS in treating a variety of psychiatric and neurological disorders, the mechanisms underpinning its therapeutic efficacy remain unclear (McIntyre et al., 2004; Ashkan et al., 2017). […] We hypothesise that these mechanisms would reduce the pathologically increased firing rate in the STN in PD (and thus the pain symptoms), as well as nociceptive responses.”

As mentioned in the previous review, the authors should be careful regarding statements about pain vs. noxious stimulation. The papers quoted to support "pain can modulate background activity" involve the responses to somatic stimulation, not the presence of a pain percept. Attention to the pain – nociceptive distinction should also be applied to the whole section starting with "Therefore, the main objective of the present work…" (other examples in subsection “Nociceptive responses in afferent structures: SC and PBN”, subsection “STN nociceptive responses in a rat model of Parkinson’s disease”, last paragraph, Discussion, first and fourth paragraphs, and many more places).

We apologize for the remaining misuse of the word pain. An attentive reading of the manuscript has been done, and ‘noxious stimulation’ and ‘nociception’ have been substituted where appropriate.

"we could differentiate three types of STN neurons, which either showed an increase, a decrease or no change of their baseline firing rate with the introduction of the noxious stimulation. The spontaneous firing rate of STN "down" cells was significantly higher compared to the two other groups, suggesting the possibility of a separate group of cells." I have two comments about this. First, the three "types" identified refer to the three possibilities. This is hardly a surprising finding unless the three types of responses are correlated with an independent metric. My second comment is that another possibility regarding the higher FR in down cells is that other cells may also be inhibited when their baseline FR increases which thereby allows a decrease to be detected.

As indicated in subsections “Baseline firing rate” and “Statistics”, we performed an individual analysis on each of the 98 STN cells to test if the change of their baseline firing rate with the introduction of the stimulation was statistically robust or not. Based on the results of the analysis, we could determine if the change of the baseline firing rate reached statistical significance or not and if the significant change was an increase or a decrease. Hence, although the fact that the FR changes is not necessarily surprising, the magnitude of that change is. In addition, although not exactly an *independent* metric, the higher baseline FR in the down neurons does suggest that the firing rate changes are more than simple statistical variations in a single population.

Regarding the second comment, as mentioned in the Discussion, further work is indeed required to elucidate the connectivity of STN subpopulations underlying their change of baseline with the noxious stimulation.

It is entirely possible that the higher firing rate of the down neurons leads to inhibition in other neurons in the STN. Many of the glutamatergic STN projection neurons have intranuclear (local) axon collaterals making intrinsic, presumed excitatory connections within the STN (Kita, Chang and Kitai, 1983), and there is evidence for inhibitory interneurons in the STN – although the issue is somewhat controversial (Iwahori N, Yelnik and Percheron, 1979; Afsharpour, 1985; Pearson et al., 1985; Rafols and Fox, 1976). A significant number of STN neurons expressing GAD and morphologically different from STN principal glutamatergic neurons has been described (Nisbet, 1996; Levesque and Parent, 2005). Because of the GABAergic nature of those small cells, they are thought to represent subthalamic interneurons, while caution has been raised by Nisbet and collaborators that those GABA cells may also represent GABA-containing subthalamopallidal projection pathways. Both studies quantified the proportion of those neurons between 5.0 and 7.5% of STN total population. However small, those putative interneurons may indeed contribute to the decrease of baseline FR in other cell populations as well as the decrease in FR in the down cells following the activation of the up cells under noxious stimulation.